



# EARLS: A runoff reconstruction dataset for Europe

Daniel Klotz[1,5], Peter Miersch[1], Thiago V. M. do Nascimento[2,3], Fabrizio Fenicia[2], Martin Gauch[4], and Jakob Zscheischler[1,6,7]

[1]Department of Compound Environmental Risks, Helmholtz Centre for Environmental Research — UFZ, Leipzig, Germany
[2]Eawag: Swiss Federal Institute of Aquatic Science and Technology, Dübendorf, Switzerland
[3]Department of Geography, University of Zurich, Zurich, Switzerland
[4]Google Research, Zurich, Switzerland
[5]Google Research, Munich, Germany
[6]Department of Hydro Sciences, TUD Dresden University of Technology, Dresden, Germany
[7]Center for Scalable Data Analytics and Artificial Intelligence (ScaDS.AI), Dresden/Leipzig, Germany

**Correspondence:** Daniel Klotz (daniel.klotz@ufz.de)

**Abstract.** Data drives our understanding of hydrological processes, supports model development, and enables anticipatory water management. This contribution introduces EARLS: European Aggregated Reconstructions for Large-sample Studies. EARLS offers daily streamflow reconstructions for more than 10,000 basins in Europe including uncertainty estimates, covering the period from 1953 to 2023. The reconstruction is derived from a single Long Short-Term Memory (LSTM) based rainfall–runoff model trained on more than 5,000 basins. LSTMs represent the state of the art in rainfall–runoff modeling and are well suited to provide predictions in ungauged basins. We evaluate the quality of the reconstruction through quantitative evaluation on two held-out sets of basins and by conducting a qualitative assessment that compares EARLS-based peak flows and flood timing to previous large-scale hydrological studies. EARLS represents a new generation of datasets that harness the capabilities of Deep Learning to obtain accurate and high-resolution data. EARLS is available at https://doi.org/10.5281/zenodo.13864843 (Klotz et al., 2024b).

## 1 Introduction

Data availability is central to hydrological science. It is the basis for advancing our understanding of hydrological processes, building prediction models, and anticipatory water management. However, in many regions and periods, observations are scarce. Practitioners often rely on hydrological model predictions to compensate for these deficiencies (e.g., Beck et al., 2015; Do et al., 2018; Ghiggi et al., 2019; O. and Orth, 2021). The resulting dataset are referred to as reconstructions. More generally, we can think of reconstructed data as datasets that are generated from observational records with the aid of models to address gaps in the data. There exist many challenges associated with making reconstructions. In particular, the non-linearity between drivers and the unique characteristics of each basin make it difficult to use process-based hydrological models for high-quality reconstructions (e.g., Addor et al., 2018). Here, purely data-driven approaches provide a great opportunity. Current approaches based on Machine Learning (ML) are able to simulate a diverse range of rainfall–runoff responses when trained using data from multiple basins (Kratzert et al., 2019b, 2024). They are able supply high-quality predictions (e.g., Kratzert et al., 2019b;





Mai et al., 2022; Jiang et al., 2022, 2024) and are suitable for simulating even in ungauged basins (Kratzert et al., 2019a; Nevo et al., 2022; Nearing et al., 2024). These characteristics render data-driven approaches an exceptional but currently underused tool for large-scale hydrological analyses.

Data availability also plays a particularly important role in Large Sample Hydrology (LSH). LSH concentrates on multiple basins, as opposed to conducting detailed studies in individual regions. A central question in LSH is how we can transfer knowledge about runoff responses between different basins, given their hydrological similarities (e.g., Hrachowitz et al., 2013; Peters-Lidard et al., 2017). Large-scale datasets allow us to identify patterns and formulate conclusions about hydrological processes across diverse regions. Recent advancements in LSH have led to the development of several large-scale hydrological

databases, compiling streamflow records from multiple locations. Notable global databases are: (1) The Global Streamflow Indices and Metadata Archive (GSIM; Do et al., 2018; Gudmundsson et al., 2018), which contains monthly, seasonal and yearly indices for over 35,000 locations; (2) The Global Runoff Data Center database (GRDC; Bundesanstalt für Gewässerkunde (BfG), 2023), which provides discharge estimates for over 10,000 locations; and (3) The Caravan project (Kratzert et al., 2023; Färber et al., 2023), which incorporates already open-source published streamflow data from various countries.

The requirement to drive rainfall–runoff models with meteorological forcings led to the development of integrated datasets that include meteorological time series such as precipitation and temperature. To our knowledge, the Model Parameter Estimation Experiment (MOPEX Duan et al., 2006) provided the first openly available large sample dataset of this kind. It contains 431 basins within the United States. Some of the most important contributions in popularizing large-scale datasets after MOPEX stem from the basin Attributes and MEteorology for Large-sample Studies (CAMELS) initiatives (Addor et al., 2017; Alvarez-

Garreton et al., 2018; Coxon et al., 2020; Chagas et al., 2020; Fowler et al., 2021; Höge et al., 2023; Loritz et al., 2024), and other derivations such as LamaH (Klingler et al., 2021; Helgason and Nijssen, 2024) and CABra (Almagro et al., 2021). Each CAMELS dataset is tailored to a specific region, but at its core follows the same logic—connecting meteorological variables and static basin attributes with streamflow data. The Caravan project (Kratzert et al., 2023) incorporates already open-source published streamflow data from various CAMELS countries — and with its recent extension also contains the sharable GRDC

data (Färber et al., 2023). In this study, we however use EStreams (do Nascimento et al., 2024) as our "raw material" for model building. EStreams also constitutes an integrated dataset, but goes into a different direction: It provides basic data for setting up hydrological models, but also a catalog streamflow data from national data providers. Unlike the CAMELS and Caravan datasets, it focuses only on the European scale. It also offers higher spatial resolution, and is designed to be continuously updated, providing the latest records in both time and space.

Despite these numerous developments in building large-scale databases for hydrology, important sampling gaps exist. This holds in particular with respect to high-quality runoff observations. This limits the usability of said databases for certain scientific applications and decision-making processes at a pan-European level. To address these limitations, here we present a data-driven daily runoff reconstruction product for natural streamflow. We name it EARLS: European aggregated reconstruction for large-sample studies. Our main goal for EARLS is to provide data-driven streamflow reconstructions that enable the

analysis of hydrological processes in the style of Blöschl et al. (2017). The reconstructions represent daily simulations of natural streamflow, are provided in mm, and cover the period from 1953 to 2020.





One can view EARLS as part of a new generation of datasets that leverage ML to achieve highly accurate predictions (e.g., O. and Orth, 2021; O et al., 2022; Nasreen et al., 2022; Kraft et al., 2024). We employ a rainfall–runoff model based on Long Short-Term Memory (LSTM; Hochreiter and Schmidhuber, 1997) to create these reconstructions. Recent studies have demonstrated

the accuracy of this approach in various contexts (e.g, Kratzert et al., 2019b, a; Mai et al., 2022; Nearing et al., 2024). The model incorporates static attributes that describe basin properties (say, average elevation) and a series of meteorological forcings (say, daily precipitation) to simulate streamflow for a given basin, following the approach introduced by Kratzert et al. (2019b) and Klotz et al. (2022). We evaluate the resulting simulations in terms of predictive performance for ungauged basins. Like EStreams, EARLS focuses on Europe. In particular, we use a subset of the EStreams basins with CAMELS attributes for

training our model (see Sect. 2.1), aiming to provide spatially extensive long-term streamflow reconstructions with uncertainty estimates, including for ungauged basins across Europe. In order to showcase the value of our dataset, we compare the long-term flood timing and changes in annual maxima from EARLS against observation-based estimates from Blöschl et al. (2017) and Blöschl et al. (2019).

## 2 Method

EARLS provides streamflow reconstructions for 17,043 European basins. The data are in daily resolution, comprise uncertainty estimates, and span for each basin from January 1, 1953, to June 30, 2023. The time period is constrained by the meteorological forcing used (see below). We plan to extend it later as dataset gets updated. Gaps in the reconstructions only occur when the dynamic inputs are erroneous — which can happen if the meteorological forcing have gaps at different timesteps. EARLS contains 14,161 basins without data gaps, 2,655 with gaps of variable lengths, and we were not able to produce a simulation

for 227. We also keep these in EARLS since the meteorological forcing is regularly revised and we plan to update on a regular basis (Sect. 4). Since it is not appropriate for royalty to be alone, we plan to produce EARLS versions with less and less gaps in the future, and eventually even start sister projects with different focal points. The basins for model training are from EStreams (Appendix A1), but we also derive a set of virtual basins to provide continuous spatial coverage (Appendix A1). The model is lumped (which necessitates an aggregation of the inputs at the basin level) and provides not only point estimates but also

uncertainty estimates in the form of a distribution prediction (Sect. 2.2).

### 2.1 Training and evaluation data

We spatially aggregate dynamic inputs (i.e., meteorological forcings) and static inputs (i.e., basin-specific static attributes) for each basin. The basin shapes are either derived from EStreams (do Nascimento et al., 2024) or HydroATLAS (Linke et al., 2019). All streamflow data is in millimeters per day (mm day$^{-1}$). We use 4 dynamic inputs (i.e., precipitation, minimum,

maximum, and mean temperature) that we aggregate basin wise from version 28 of E-OBS dataset (Klein Tank et al., 2002; Haylock et al., 2008). E-OBS is a daily-resolution gridded dataset covering the European region (25N-71.5N x 25W-45E). This defines the extent of EARLS in both space and time. E-OBS interpolates station data from European National Meteorological Services and other providers, spanning 1 January 1950 to present. It comprises time series of meteorological variables such as

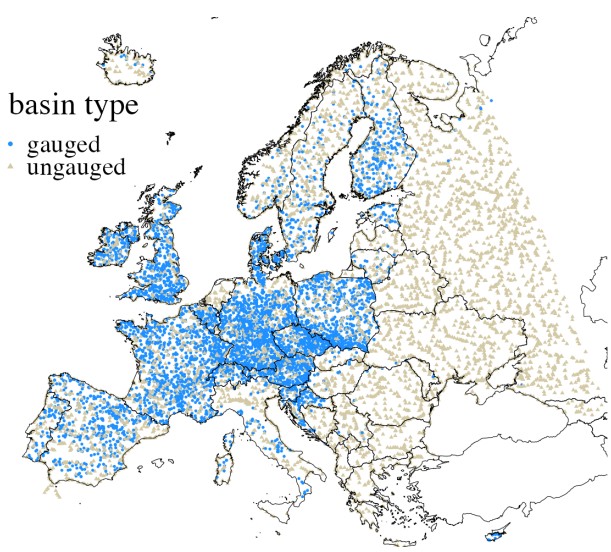

**Figure 1.** Spatial distribution of the gauged (blue circles) and ungauged basins (grey triangles).

daily mean, maximum, and minimum temperature, daily total precipitation and mean sea level pressure. For the static inputs,
we use 13 attributes that we aggregate from HydroATLAS, namely: basin area, the average elevation, the average slopes, the
average stream gradient, the average long-term air temperature, the minimum long-term air temperature, the maximum long-
term air temperature, a global aridity index (Zomer et al., 2022), a global climate moisture index (Hijmans et al., 2005), the
average fraction of sand the average fraction of clay, the average fraction of silt, and the average organic carbon content.

### 2.2  Rainfall–runoff model

Our LSTM-based rainfall–runoff model uses dynamic and static inputs to estimate streamflow. Since the LSTM is a deep
learning architecture we will use concepts and language from machine learning to describe how we set it up. For example, for
the model selection we will use a triple split (training, validation, and test set) and we will refer to the selection procedure as
training (and not, say, as model calibration as is usual in hydrology). To provide uncertainty estimates, we adapt a simplified
version of the approach from Klotz et al. (2022). In short, instead of estimating the streamflow directly, the LSTM outputs
the three parameters of an asymmetric Laplacian distribution (a double exponential with a parameter) — and is trained using
maximum likelihood (Fig. 2). From here on out we refer to this model as the EARLS LSTM. The next two sections describe
how we set up the training and evaluation of the EARLS LSTM. Further technical details are available in Appendix A.

### 2.3  Evaluation

We use a mix of quantitative and qualitative measures to evaluate the quality of EARLS. We check the model performance
for the test basins (Sect. 2.3.1), examine whether model performance is related to static or dynamic basin characteristics (Sect.

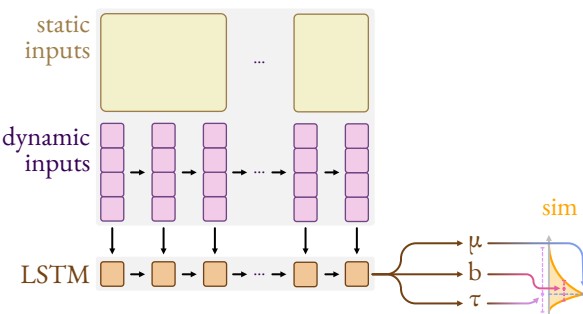

**Figure 2.** High-level model conceptualization. The LSTM outputs the location parameters $\mu$, the scale parameter $b$, and the asymmetry parameter $\tau$ to parameterize an asymmetric Laplace distribution for each predicted timestep. For the use and definition of the static and dynamic inputs we refer to Appendix A1 and Appendix A2.

2.3.2), perform a comparative analysis with a process based model on the basis of 161 separately chosen basins (Sect. 2.3.3), and conduct a qualitative assessment against published literature results (Sect. 2.3.4).

### 2.3.1 Model performance

For the model evaluation performance we report the performances for the training, validation and test set (for a technical
definition we refer to Appendix A3). It normally does not make sense to report training and validation performances in a ML context, since one is generally interested in generalization — and performance measures are biased for the training and validation sets. However, since we also publish simulations for these basins we argue that it is informative for users to also report the respective performances. Specifically, we report Nash–Sutcliffe efficiency (NSE) for each basin and over the time horizon of a given portion. For a given basin it is defined as:

$$\mathrm{NSE} = 1 - \frac{\sum_{t=1}^{T}(o_t - s_t)^2}{\sum_{t=1}^{T}(o_t - \bar{o})^2}, \tag{1}$$

where $t = 1, 2, ..., T$ is the time index for the given basin and time horizon, $o$ the observations, $s$ the simulations, and $\bar{o} = 1/T \sum_{t=1}^{T} o_t$ the sample mean of the evaluated data. Appendix C also reports the model performance on the test set with regard to other metrics.

### 2.3.2 Post hoc examination

As a post hoc analysis of the model evaluation we investigate the relationship between model performance and (a) static inputs or (b) streamflow respectively. For this, we use the 500 basins from the test set (Appendix A3) and the basin similarity measure from Bertola et al. (2023). In their conception $\mathcal{Z} = Z_1, Z_2, ..., Z_M$ is a collection of different basin attributes where each $Z_M = [z_1^{(m)}, z_2^{(m)}, ..., z_N^{(m)}]$ is a vector of a given attribute $m$ (e.g., average elevation) over the different basins $M$. That is, each entry in the vector is property of interest (e.g., the average elevation) for a given basin. Then, for two basins $i$ and $j$ the




"Bertola distance" $d$ is the Euclidean distance:

$$d(\mathcal{Z}, i, j) = \sqrt{\sum_{m=1}^{M} \left( \frac{z_i^{(m)} - z_j^{(m)}}{sd(Z_m)} \right)}, \tag{2}$$

where $\mathrm{sd}(Z_n)$ denotes the standard deviation of $Z_n$. We make use of two different choices for $\mathcal{Z}$. The first set consists of the 13 static inputs of our model and the second choice depends on the streamflow only. For the latter, we choose the logarithm of the mean of the annual maximum specific streamflow following Bertola et al. (2023). That is, the annual maximum discharge normalized to a basin area of $100\mathrm{km}^2$ (see Bertola et al., 2023). Formally, we can express these choices as

$$d_a(\mathcal{Z} = S_1, S_2, ..., S_{13}, i, j) = \sqrt{\sum_{m=1}^{M} \left( \frac{s_i^{(m)} - s_j^{(m)}}{sd(S_m)} \right)}, \tag{3}$$

and

$$d_b(\mathcal{Z} = \log(\widetilde{Q}), i, j) = \sqrt{\sum_{m=1}^{M} \left( \frac{\log(\widetilde{q}_i^{(m)}) - \log(\widetilde{q}_j^{(m)})}{\mathrm{sd}(\log(\widetilde{Q}_m))} \right)}, \tag{4}$$

where $S_i$ refers to the static inputs, and $\log(\widetilde{Q})$ refers to the normalized annual discharges. The distance $d_a$ does solely depend on the static inputs and is therefore always computeable. In contrast, $d_b$ does only depend on the runoff and can hence exclusively be used for model diagnosis in gauged basins. To understand whether particularly low or high model performance might be related to static basin attributes or runoff dynamics, we analyze two subsets of basins: basins for which $\mathrm{NSE} < 0$ and those for which $\mathrm{NSE} > 0.8$. We then compare the following sets of average distances

$$D_{<0.0} = E_i(d(\mathcal{Z}, i, j))|i \neq j \quad \text{and} \quad \mathrm{NSE}_i < 0.0 \quad \forall i, j \in 1, 2, ..., M, \tag{5}$$

and

$$D_{>0.8} = E_i(d(\mathcal{Z}, i, j))|i \neq j \quad \text{and} \quad \mathrm{NSE}_i > 0.8 \quad \forall i, j \in 1, 2, ..., M. \tag{6}$$

If there is a pattern between within the two subsets $D_{<0.0}$ and $D_{>0.8}$, then it could be possible to predict whether the model performs well or not — and modelers can use that information to infer whether a model performs well or not for a given basin.

### 2.3.3 Model comparison

We use the mesoscale hydrological model (mHM; Samaniego et al., 2010; Kumar et al., 2013) as a reference model for the comparative evaluation. Specifically, we run mHM in two configurations for 161 seperately delineated gauged basins. The first configuration is represented by the global parametrization from Kumar et al. (2013), hereafter referred to as "mHM default". For the second configuration, we calibrate the parameters for each basin with 1970-1999 for training and use the years 2000-2020 for testing, hereafter referred to as "local mHM". In both cases, we use E-OBS for the dynamical inputs (precipitation, average temperature and potential evapotranspiration estimated with the Hargreaves-Samani method (H. Hargreaves and A. Samani,





1985) and static inputs from Tab. 1. Since mHM is intrinsically a semi-distributed model, we set the spatial resolution of the model to 0.25 degree. For the local mHM, we maximize the NSE using the dynamically dimensioned search algorithm (Tolson and Shoemaker, 2007) with 1000 iterations. The comparative evaluation with the EARLS LSTM is therefore asymmetric: Firstly, the reconstructions are tested for an ungauged setting, while local mHM — as our reference model — is calibrated using a traditional time-split setting. Secondly, the mHM default is not a result of a specific calibration process for the task at hand and is hence disadvantaged.

**Table 1.** Morphological data used for the hydrological model mHM.

| Description | Source |
| --- | --- |
| Digital Elevation Model from U.S. Geological Survey (USGS) | Danielson and Gesch (2011) |
| Soil map from SoilGrids | Hengl et al. (2017) |
| Land cover from the European Space Agency (ESA) | Arino et al. (2012) |
| LAI climatology from NASA Global Inventory, Monitoring, and Modelling Studies | Tucker et al. (2005) |

### 2.3.4 Qualitative assessment against published literature

We use EARLS to redo the core parts of the flood-timing analysis from Blöschl et al. (2017) and the flood-peak trends analysis from Blöschl et al. (2019). Both studies use observations from 1960 to 2010 (albeit the full period is not available for all gauges). We select the same timeframe and use the publicly available code from Blöschl et al. (2019) to compute the trends and spatial interpolations of the peak trends. To recreate the results of Blöschl et al. (2017) we adapt it for the flood-timing analysis according to their supplementary material (Appendix D provides details of the adoption process).

Our goal with these reproductions is to provide a visual check for the data quality of EARLS. However, as far as we know, this also constitutes the first corroboration of the results from Blöschl et al. (2017) and Blöschl et al. (2019) with different raw data (since the underlying raw data from the original papers are not available open access).

## 3 Evaluation results and discussion

In the following, we show the results of our model evaluation. The structure follows the exposition from Sect. 2.3.

### 3.1 Model performance

The EARLS LSTM achieves a median NSE of 0.66 for the 500 test basins. We view this as a good result, given our splitting strategy leads to potentially difficult to predict ungauged basins for the evaluation (Sect. 2.1) and we only use a single dynamic input product (as, for example, opposed to Kratzert et al., 2021). Still 10% of the basins exhibit NSE values that are lower than 0.0 (Fig. 3). These results are similar to the ones in Kratzert et al. (2019a), but worse than the ones from Mai et al. (2022).


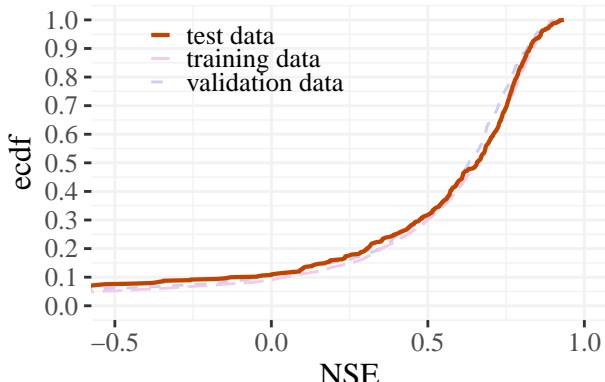

**Figure 3.** Empirical cumulative density function of the Nash–Sutcliffe efficiency (NSE). Each point on the line represents the model performance for one of the 500 test basins. The red-solid line shows the performance for the test data. The dashed lines show the performance for the training and validation data.

The best basin has an NSE of 0.93 and the worst of -25.42. These values are not randomly distributed (Fig. 4). Appendix C provides empirical cumulative distribution functions for other metrics.

We posit that most low accuracy values can be attributed to data quality issues rather than to shortcomings in the LSTM. First, we would like to point out that there is evidence that the NSE can be rather erratic in arid climatic regimes (e.g., Duc and Sawada, 2023; Klotz et al., 2024a). However, more concretely there exist hydrological reasons that might explain the bad performances. Specifically, many of the low accuracy basins (a) are influenced by human activities, such as the presence of dams and reservoirs (Portugal and Spain); (b) exhibit extensive canal systems and numerous lakes (Denmark, Sweden, and Norway); (c) contain karstic geology (Central Europe); or (d) are situated regions where the meteorological forcing data are scarce (Iberian Peninsula and southern Italy; see Do et al., 2018). In our data preprocessing we filter out basins that exhibit a high degree of human influence (Appendix A1). In Spain and Portugal, anthropization is primarily caused by numerous dams constructed for water supply. The identification of such structures at such a large-scale is challenging (Senent-Aparicio et al., 2024). Hence, EStreams may not correctly capture the total number of dams and reservoirs in many regions because of the used data sources (as discussed in Salwey et al., 2024). Furthermore, the high number of natural lakes and the presence of canalization systems may also negatively influence the model's performance in these areas. The same applies for the presence of karstic systems, which poses a challenge for closing the water balance in some basins. EStreams made significant efforts to label such basins do Nascimento et al. (2024). However, despite these efforts we where not able to produce ex-ante labels for said basins or create an adequate criterion to filter them out (Appendix A1). This affects both, model training and evaluation, since some signals are not learnable in the first place.

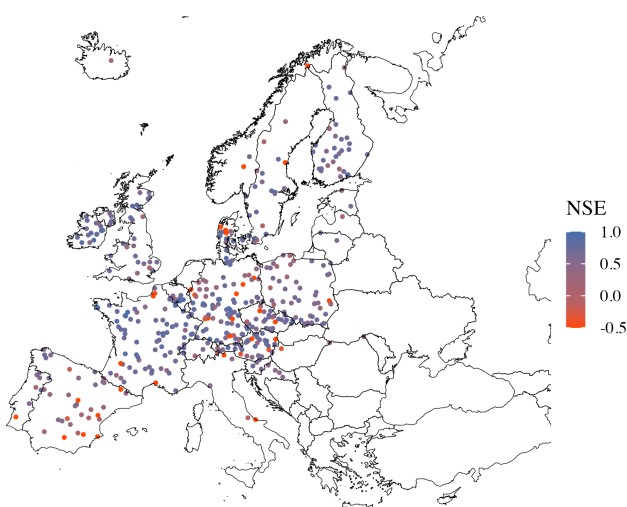

**Figure 4.** Spatial distribution of the NSE values for the test evaluation. The colors mirror Fig. 3: NSE values close to the lower bound of -0.5 or below are red and the closer the values are to the upper bound (i.e., 1) the more blue the coloring.

## 3.2 Post hoc examination

We observe a shift between badly performing basins in $D_{<0,0}$ and well performing basins in $D_{>0.8}$ (Fig. 5a). This shift is absent for the same analysis using static attributes (Fig. 5b). In fact, for the latter, the mode of the distribution is at a lower value despite comparing higher dimensional entities (i.e., the 13 static attributes). These quantitative results suggest that it is easier to discriminate the model performances with streamflow observation than with static attributes. The results also align with our hydrological justifications for model performance (Sect. 3.1), since anthropogenic factors are not encoded in static attributes but are reflected in the streamflow.

## 3.3 Model comparison

Tn terms of performance, the EARLS LSTM ranks between the mHM default and the locally calibrated mHM (Fig. 6). That is, until approximately the 15th percentile of NSE values, the EARLS LSTM performance is close to the mHM default performance, and roughly starting at the 60th percentile, it is close to the local mHM model. For the remaining 15% it is somewhere in between, and for the best performing basins EARLS LSTM even outperforms the latter. All in all, we argue that these are promising results for an ungauged evaluation — especially if we keep in mind that this is an asymmetric comparison: the EARLS LSTM operates "out-of-sample" (validation in space), while mHM operates "in-sample" in space (validation in time).

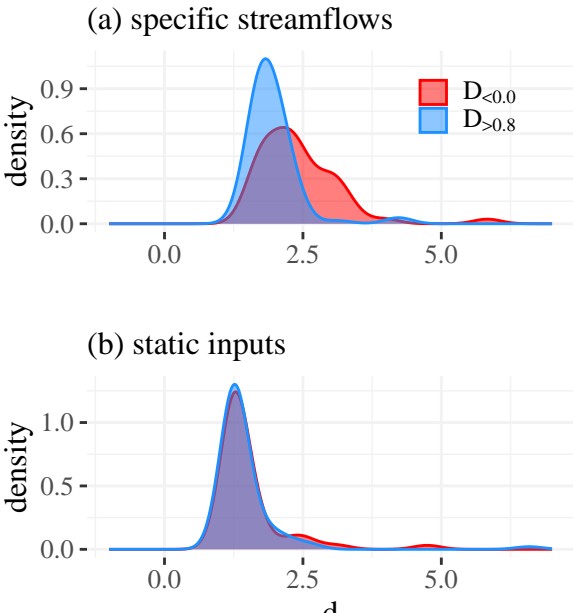

**Figure 5.** Comparison of the distributions of average "Bertola distances" (Sect. 2.3.2) for the subset of basins with low NSE values ($D_{<0.0}$) and the subset of basins with high NSE values ($D_{<0.8}$).

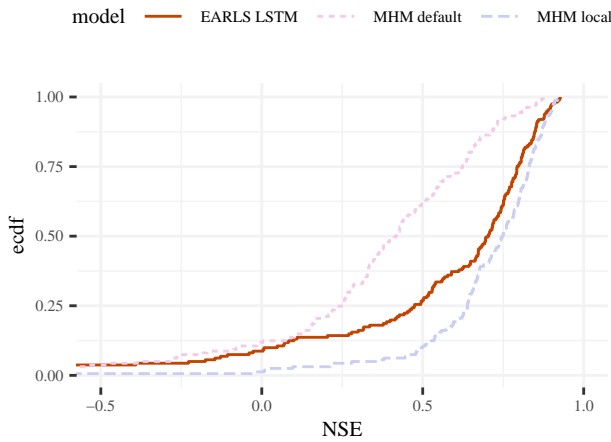

**Figure 6.** Empirical cumulative density functions for the comparative evaluation. The mHM default is a "best-guess" mHM calibration that summarizes many studies; the mHM local are per basin calibrated models evaluated in a traditional time-split fashion; and the EARLS LSTM represents the ungauged performance of the EARLS model.

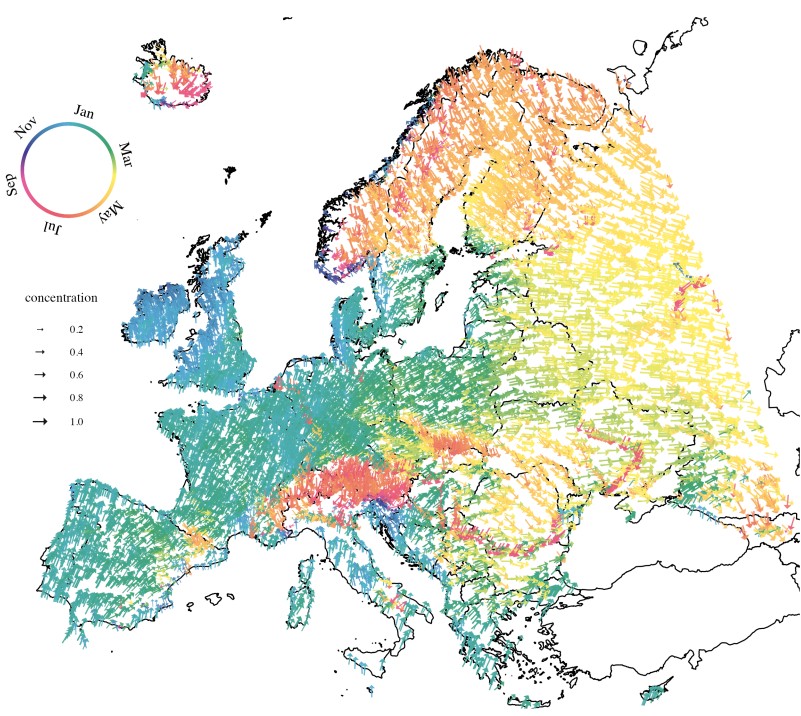

**Figure 7.** Reproduction of Fig. 3 from Blöschl et al. (2017) with EARLS data. Each arrow represents a basin outlet (either from a gauged or ungauged basin). Color and arrow direction indicate the average timing of floods over the period 1960–2010 (namely: light blue are winter floods; green to yellow are spring floods; orange to red are summer floods; and purple to dark blue are autumn floods). The lengths of the arrows indicate the concentration of floods (0, evenly distributed; 1, all floods occur on the same date).

## 3.4 Qualitative assessment against published literature

The first part of our qualitative assessment revolves around the timing of river floods in Europe. These results corresponds to Fig. 1 and Fig. 3 of Blöschl et al. (2017). We encourage readers to compare our version with these depictions since all key patterns from the original analysis are preserved in the EARLS version. Figure 7 shows the average timing of the yearly streamflow maxima from EARLS. The overall pattern of our this version closely corresponds to the original — but with a larger number of support points and a wider area of analysis. The reproduction of the corresponding analysis of flood timing trends from EARLS (Fig. 8), also mirrors the large-scale trends from Fig. 1 from the original publication. The effect of the Pyrenees, the Alps and the Carpathians are clearly visible. The 4 approximate key region with distinct drivers that Blöschl et al. (2017) highlight are also reflected in the EARLS version: (1) Northeastern Europe with earlier snowmelt; (2) the North Sea region with later winter storms; (3) Western Europe along the Atlantic coast with earlier soil moisture maxima; and (4) Parts of the Mediterranean coast (West Spain, South France, Croatia, etc.) with stronger Atlantic influence in winter.

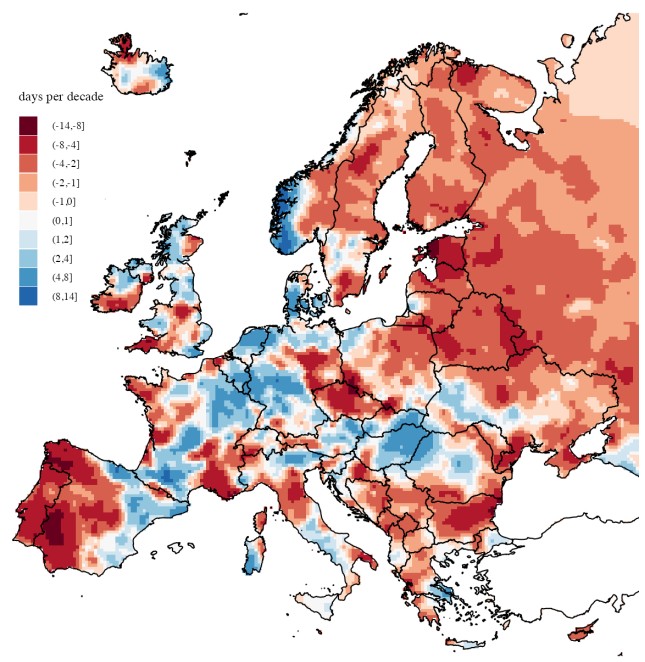

**Figure 8.** Our EARLS-based remake of the flood peak timing trends from Blöschl et al. (2017). Positive trends are depicted in red, negative trends in blue. The most extreme negative trends can be spotted in Spain/Portugal and the strongest positive trends in the west of Sweden. In general, the overall patterns match the ones from the original publication, but show more detail.

The plotting data from Blöschl et al. (2019) are openly available. We can therefore directly compare them with our EARLS version. To this end, we made a reproduction of their results (Fig. 9a) and contrasted it with the corresponding EARLS version (Fig. 9b). Both show similar large-scale trends, but significant discrepancies exist for large and small-scale patterns. Scandinavia and North-East Europe have the biggest divergence in terms of large-scale patterns. There, the original version shows
slightly positive trends, while the EARLS version shows no or slightly negative trends. In absolute terms the differences are not that large, but the extent at which the differences occur is noteworthy. The largest absolute differences, on the other hand, occur in the north of Portugal (where the EARLS version shows major negative trends) and West Russia/Ukraine (where the original analysis depicts substantially larger negative values).

Some differences can be explained by data availability. In general, the EARLS version appears to have more details, which,
we posit, is linked to a higher number of data points used for kriging. EARLS lacks reconstructions for the Asian part of Turkey, while Blöschl et al. (2019) have observations there. Blöschl et al. (2019) have limited observations in western Russia and northern Ukraine, showing strong negative trends in flood magnitudes. EARLS has denser coverage, showing negative trends restricted to a specific eastern region. However, E-OBS is based on few observations stations in Eastern Europe (the actual number varies from variable to variable; see, e.g., Fig. 6 in do Nascimento et al., 2024). Neither of the two analyses
have data in the most northern part of West Russia, resulting in flat spatial trends. Furthermore, in the data from Blöschl et al.



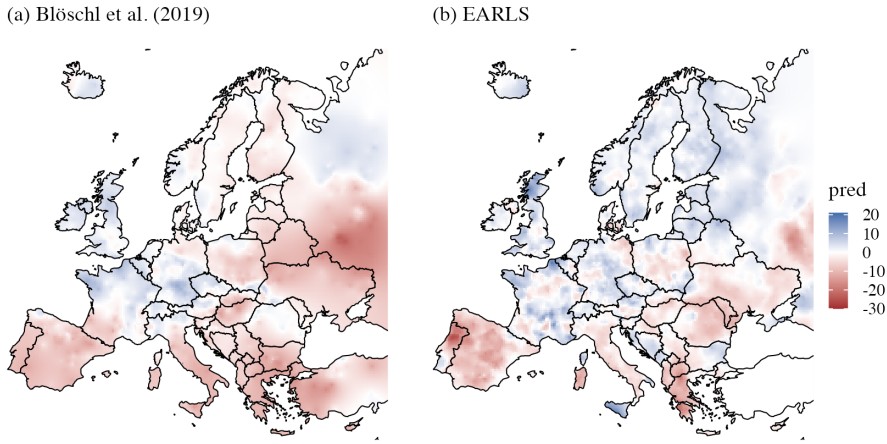

**Figure 9.** Comparison of the trend analysis for flood peaks from 1960 to 2010 derived from different datasets: Our reproduction with data of annual maxima from (a) Blöschl et al. (2019), and (b) EARLS. Positive trends are in blue, negative ones in red. Blöschl et al. (2019) manually binned their data into specific classes. In contrast, we choose a continuous color-scale centered around zero.

(2019) the availability of streamflow observations varies widely from station to station, while EARLS is more homogeneous in this regard, potentially leading to different trend estimates. In summary, we argue that the results of our qualitative assessment show the merit our the EARLS data for scientific inquiry and corroborate the quality of the simulations.

## 3.5 Limitations

Our results suggest that EARLS as a dataset is well suited for large-sample hydrological studies in Europe. However, there exist several important limits to EARLS: The simulation quality is restricted by (a) the streamflow observation quality, (b) the input quality, and (c) the capabilities of the EARLS LSTM. The quality of streamflow and input data varies in space (e.g., different measurement standards across countries) and time (e.g., improvements in measurement technique over time). As a matter of fact, some of the observational signals are highly atypical and it is likely that they can generally not be modeled

with the available information (Sect. 3.1). The same kind of reasoning applies to the forcings. For example, E-OBS is more accurate in high station-density regions like Germany and Austria, and less so in lower density areas like Spain, Portugal, or Eastern Europe (see do Nascimento et al., 2024). These results suggests the existence of non-trivial distribution shifts in the reconstructions. From a ML perspective distribution shifts are not trivial. Lastly, with regard to (c), a machine learning model — such as the EARLS LSTM — by design, can only capture the signal that is in the data. We chose an LSTM-based approach

since it represents the best simulations in gauged and ungauged settings (Kratzert et al., 2019b, a; Mai et al., 2022; Nearing et al., 2024). Nonetheless, we kept the setup simple for ease of use. For instance, we use a limited set of dynamic inputs that are available in many observation-based datasets and use a simple single Laplacian distribution in the output and a single



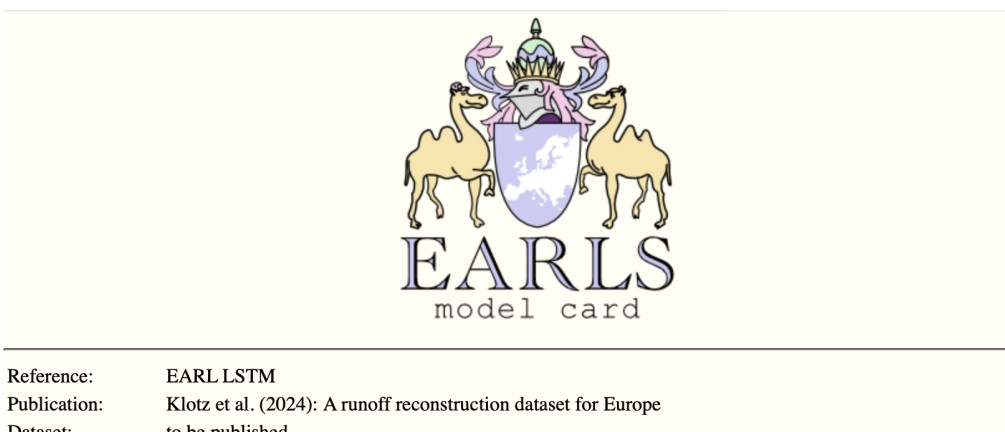

| Reference: | EARL LSTM |
| --- | --- |
| Publication: | Klotz et al. (2024): A runoff reconstruction dataset for Europe |
| Dataset: | to be published |
| Name (contact): | Daniel Klotz (daniel.klotz@ufz.de) |
| Model type: | LSTM |
| Streamflow source: | Based on data references from EStreams (do Nascimento et al., 2024). |
| Dynamic inputs | |
| 1. Source: | E-OBS (Klein Tank et al., 2002; Haylock et al., 2008). |
| 2. Inputs: | Daily precipitation and mean temperature. |
| 3. Comment: | Aggregated at basin level. |
| Static inputs | |
| 1. Source: | HydroATLAS (Linke et al., 2019). |

**Figure 10.** First part of the model card for the EARLS LSTM.

meteorological product for the dynamic inputs. We did not conduct extensive intercomparisons and expect better results with more complex setups. In that case, the current EARLS version will serve as a (strong) baseline.

Further, EARLS incorporates distributional prediction for each time step by providing the conditional parameters of an asymmetric Laplace distribution (Sect. 2.2). This does not represent the full uncertainty of the prediction. On top of that, the conditional distribution assumes no knowledge of the streamflow. Thus, if we naively take samples at a given time step, then this does not account for the autocorrelative nature of the streamflow.

## 4    Data and code availability

The project homepage for EARLS is https://earls-dataset.github.io/. We envision the page as a living document for future news and updates. The EARLS data is available at https://doi.org/10.5281/zenodo.13864843 (Klotz et al., 2024b). It includes streamflow reconstructions for 17,043 European basins. We build the dataset with extensibility in mind. The idea here is that the current structure becomes a blueprint for potential future expansions. Specifically, the data of EARLS is organized as follows:

- The "coordinates.csv" file contains basin outlet information with 4 columns: basin id (idx), type, and estimated latitude
(lat) and longitude (lon) of the outlet.

- The "license.md" file contains information about the licensing.





- The "shapefile" folder includes a shapefile with all basin boundaries.

- The "reconstructions" folder contains CSV files. Each file is named after the basin id and has at least two columns: date and simulation. The simulations are given in mm day$^{-1}$. Additional columns can be used to provide more information. For the current EARLS we added two additional columns that provide the remaining parameters for the uncertainty estimation (Sect. 2.4).

- The "model-card" folder contains 3 files: "model-card.html", and "earls-crest.png". The html document includes the png as logo and renders a model card (Fig. 10). A model card is a short summary of the model genesis, designed to increase transparency by communicating information about trained models to broad audiences (Mitchell et al., 2019). We include all three files in the dataset so that future extensions can adapt them with maximal ease. We will also host the markdown files on the main home so that the permanent identifier within the model card can be used to access the data from there.

- Additional data/folders are optional, but can be used to provide background information. For instance, to enable benchmarking the current EARLS version also contains an "inputs" folder, which comprises the basin-aggregated dynamic and static inputs (see Appendix A1 and Appendix A1). :

  - For the dynamic inputs (derived from Klein Tank et al., 2002) we use precipitation in mm day$^{-1}$, daily minimum temperature in °C, daily maximum temperature in °C, and daily mean temperature in °C.

  - For the static inputs (derived from Linke et al., 2019) we use basin area in km$^2$, average elevation in m, average slopes in degrees, average stream gradient in dm km$^{-1}$, average long-term air temperature in °C, minimum long-term air temperature in °C, maximum long-term air temperature in °C, a global aridity index (Zomer et al., 2022), a global climate moisture index (Hijmans et al., 2005), average fraction of sand in %, average fraction of clay in %, average fraction of silt in %, and average organic carbon content in t ha$^{-1}$.

We provide the code for the EARLS-LSTM, our experiments, and plots at https://github.com/earls-dataset/paper-code (in addition, a snapshot of the code will be provided at a suitable repository after the review process).

## 5 Conclusions

We provide a data-driven streamflow reconstruction product for Europe, called EARLS (European aggregated runoff reconstruction for large-sample studies). The main purpose of EARLS is to enable large-sample hydrological streamflow analysis at European scale. As of now, EARLS consists of reconstruction for 17,043 European basins from 1953 to 2023, at a daily scale. From these, over 11 thousand represent ungauged basins from HydroATLAS. This motivates our model-driven approach for the reconstructions. The model also enables us to provide predictions that are distributional in nature. That is, for each time step EARLS provides a conditional uncertainty estimate — which can, for example, be used to compute the likelihood of a given model. This, for example, enables researcher to explore which situations are associated with what kind of uncertainties or to train classical models on top of it using the information in their objective functions.





As such, one can see EARLS as part of a new generation of datasets that leverage machine learning (e.g., O. and Orth, 2021; O et al., 2022; Nasreen et al., 2022; Kraft et al., 2024). Streamflow is by its nature a perhaps more virtual variable than many others (see discussion in: Beven et al., 2012). Nonetheless, using machine learning to create reconstructions for thousands of basins is a new qualitative dimension. The job of the model is here to extract the information that the meteorological signals contain about the streamflow and act as a *virtual sensor*. This is certainly not a replacement for real data — on the contrary, our ability to enable models in this way heavily depends on the availability of large amounts of diverse and highly qualitative data (Kratzert et al., 2024). However, it does allow us to bring in richer and new information into the datasets. In our specific, case, for example, this happens by using the information about the streamflow and its inherent uncertainties that we are able to extract from the meteorological signals. As such, we posit that machine learning will become an integral part of many datasets (even if the degree of its use might vary heavily from application to application). Maybe in the future entirely new forms of datasets will emerge from this practice, inheriting their own specific advantages and disadvantages. EARLS, and the many other datasets that are currently published in the same direction, will then be seen as stepping stones that paved our understanding for this new class of dataset.

## Appendix A: Technical details of the modeling process

This appendix lines out the technical aspects of the modeling process.

### A1 Gauged basins

We use EStreams to derive our gauged training basins (the blue-circled basins in Fig. 1). EStreams contains hydro-climatic variables and landscape descriptors, and references to openly available streamflow records for 17,130 European basins (do Nascimento et al., 2024). EStreams includes basin delineations, hydro-climatic signatures, and landscape attributes (topography, soils, geology, vegetation, and land cover), and gives the necessary information to access daily streamflow data from the data providers, which we cannot redistribute. The data quality of EStreams basin does, however, vary considerably. Thus, we filter out gauged basins according to the following criteria:

1. Each basin needs to have high-quality delineations (see Table 3 in do Nascimento et al., 2024).

2. To minimize aggregation errors in the basin mean attributes and simultaneously reduce the effects of channel routing, we only include basins equal or larger than $50\text{km}^2$ and smaller than $100,000\text{km}^2$.

3. We require each basin to have at least 30 years of, not necessarily consecutive, daily streamflow observations.

4. We exclude basins which, based on the attributes derived in EStreams, include more than four dams or reservoirs within the basin boundary.

5. We require the presence of meteorological time-series from E-OBS for the basins.

6. We exclude basins where hydrological signatures indicate potential data problems, based on the following criteria:



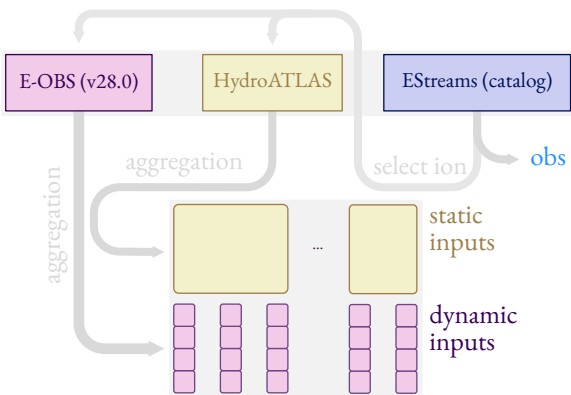

**Figure A1.** Overview of the gauged basin setup. We select the basins and observations from the EStreams catalog, derive the static inputs from HydroATLAS and the dynamic ones from E-OBS. The yellow boxes indicate that we concatenate the same static input for each timestep. The static attributes are just different from basin to basin. In contrast, the dynamic input vary for each timestep (and basin).

   (a)  The long-term average streamflow needs to be below 10 mm day$^{-1}$.

   (b)  The long-term runoff ratio (as defined in Sawicz et al., 2011) cannot be larger than 1.

325     After applying these constraints to the EStreams catalog, we are left with 5,786 basins (Fig. A1; see also Appendix A). We further partitioned these into 4,786 training basins, 500 chosen basins for validation, and 500 for testing. That is, we do not apply any time split for the training and evaluation of the EARLS LSTM. Hence, validation is carried out in space and not in time.

**A2   Ungauged basins**

330   The ungauged basins (i.e., basins without observations) add a set of virtual basins as additional support basins for EARLS. The goal here is to achieve dense coverage across Europe (gray basins in Fig. 1). We delineate the ungauged basins from the union of all upstream level 12 polygons of the level 12 layer of HydroATLAS. Like in the gauged case, the static and dynamic inputs are derived from HydroATLAS and E-OBS, respectively (Fig. A2). The resulting "simulation layer" consists of 11,277 basins (some overlapping with the gauged basins), which yields a total of 17,043 EARLS basins when combined with the gauged 335   basins that we use for training.

**A3   Training**

Ultimately, modeling for EARLS is an exercise in spatial generalization. Gauged and ungauged basins are not randomly distributed (Sect. 2.1, Fig. 1). Direct evaluation is only possible for the former, but the EARLS LSTM should also generalize to the latter. We choose a data split strategy that reflects this inherent challenge. Intuitively, our goal is to partition the data so that 340   the distribution of the training, validation, and test sets is different enough to estimate an out-of-sample model performance




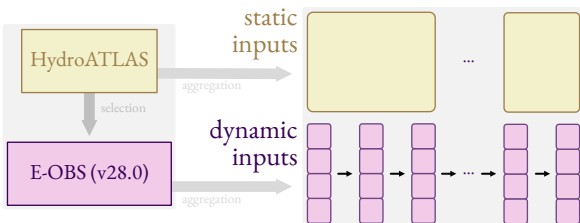

**Figure A2.** Overview of the modeling setup for the ungauged basins. We derive the basins from HydroATLAS. Like in the gauged basin setup (Appendix A1), we use the static inputs from HydroATLAS and the dynamic ones from E-OBS. The yellow boxes indicate that we concatenate the same static input for each timestep. The static attributes are just different from basin to basin. In contrast, the dynamic input vary for each timestep (and basin).

for the ungauged basins. To measure the difference in distribution, we use the Wasserstein-1 distance of the standardized static inputs (standardization is used to prevent that we just measure the differences in feature magnitudes). The Wasserstein or earth mover's distance $W_1$ is a distributional distance that measures the smallest distance between two sets of samples. It is widely used in machine learning (e.g., Arjovsky et al., 2017; Tolstikhin et al., 2017; Shen et al., 2018; Torres et al., 2021). Formally,
345 the Wasserstein-1 distance is defined as

$$W_1(\mu,\nu) = \inf_{\gamma \in \Gamma(\mu,\nu)} \int_{M \times M} |x_i - x_j| \, d\gamma(x,y), \tag{A1}$$

where $\Gamma(\mu,\nu)$ is the set of all couplings of $\mu$ and $\nu$. In theory the considered distance can be chosen, but we only consider the absolute difference here. In practice, we sample $x_i$ from a dataset $D_i$ and $x_j$ from a dataset $D_j$ respectively. The sampling is necessary, since the $W_1$ distance expects that we have the same amount of samples from the compared distributions, but we
350 use a different number of basins for each set. Namely: The training set $D_{train}$ with 5,386 basins, and validation and test sets, $D_{val}$ and $D_{test}$, with 500 basins.

We sum the distances between all pairs of the three partitions to get an overall distance $d_W$ that expresses how far apart they are from each other:

$$d_W = \frac{1}{K} \sum_k^K W_1(A_{train}^k, A_{val}) + \frac{1}{K} \sum_k^K W_1(A_{train}^k, A_{test}) + W_1(A_{test}, A_{val}). \tag{A2}$$

355 Here, $K = 20$ defines the number of repetitions, and $A_{train}^k$ indicates a random subset of size 500 from the training dataset. This is needed because $W_1$ assumes the same sample size from the distributions. To get the final split, we make create $3 \times 20$ random sets and select the partition with the largest distance $d_W$. From an optimization standpoint, obtaining large $d_W$ values is a combinatorial problem and the use of random partitioning is suboptimal. Hence, we experimented with clustering-based subsetting and approaches using an optimizer to exchange individual data points during the development. These and similar
360 strategies would align more closely with cluster-based splitting of training, test and validation sets as proposed by Mayr et al. (2018) or Sweet et al. (2023). If $d_W$ becomes too large, the training performance will not translate to the validation performance,



**Table A1.** Hyperparameter settings for the EARLS LSTM.

| Hyperparameter | Value |
|---|---|
| Initial forget gate bias | 3 |
| Hidden size | 350 |
| Batch size | 3000 |
| Number of epochs | 40 (and choose best validation performance for model) |
| Learning rate schedule (epoch:learning rate) | 0:0.001, 5:0.0005, 20: 0.0001 |
| Clipping to gradient norm | 1.0 |
| Loss | Negative Log-likelihood |
| Sequence length | 365 |
| Standard deviation of the noise added to the streamflow observations | 0.1 |

and the validation performance, in turn, will not indicate the test performance. We defer the design of better separation schemes that optimally emulate the task in question to future work.

## A4  Technical specifications

### A4.1  Hyperparameters

In order to find good hyperparameters for the EARLS LSTM we use a mixture of manual and grid search. The goal was to find a good trade-off between model simplicity and generalization. Table A1 shows the final parameters for the EARLS LSTM.

### A4.2  Distributional predictions

In order to achieve distributional predictions we adapt the approach from Klotz et al. (2022), letting the LSTM parameterize a single asymmetric Laplacian for each time step. To get the runoff estimate within the EARLS we use the location parameter of the distribution. We do the training in normalized space (as described in Kratzert et al., 2019b). For the dataset we rescale the location and scale parameters, but leave the asymmetry parameter unchanged.

## Appendix B:  Streamflow data sources for EARLS

Table B1 show the different streamflow data sources we obtained for training the EARLS LSTM (see Sect. 2).

Table B1: The EStreams derived data sources we use for training EARLS.

| Country/region | Dataset source |
|---|---|



| Austria | BML. Federal Ministry of Agriculture, Forestry, Regions and Water Management: WebGIS-Applikation 666eHYD, Wien, Austria, https://ehyd.gv.at (last access: 05 May 2023) |
|---|---|
| Belgium, Vlaanderen | VW. Vlaanderen waterinfo, Belgium. https://www.waterinfo.be/kaartencatalogus?KL=en (last access: 07 Dec 2023). |
| Belgium, Wallonie | SPW. Service public de Wallonie:L'hydrométrie en Wallonie: Observations: Debit, Belgium. https://hydrometrie.wallonie.be/home/observations/debit.html?mode=announcement (last access: 07 Dec 2023). |
| Belarus | GRDC. Global Runoff Data Center: River discharge data. Federal Institute of Hydrology, 56068 Koblenz, Germany. https://www.bafg.de/GRDC (last access: 01 May 2024 |
| Switzerland | "BAFU. Federal Office for the Environment (Switzerland). https://www.bafu.admin.ch/bafu/en/home.html (last access: 15 May 2023) |
| Höge | M. et al. CAMELS-CH: hydro-meteorological time series and landscape attributes for 331 catchments in hydrologic Switzerland. Earth Syst Sci Data 15; 5755–5784 (2023)." |
| Cyprus | GRDC. Global Runoff Data Center: River discharge data. Federal Institute of Hydrology, 56068 Koblenz, Germany. https://www.bafg.de/GRDC (last access: 01 May 2024 |
| Czechia | CHMI. Czech Hydrometeorological Institute: ISVS -Evidence množství povrchových vod. https://isvs.chmi.cz/ords/f?p=11002:HOME:5026647009329::::: (last access: 10 Jul 2023) |
| Germany, Land-Brandenburg | LBAW. Land Brandenburg Auskunftsplattform Wasser. https://apw.brandenburg.de/?th=owm_gkp/ (last access: 12 Dec 2023). |
| Germany, Baden-Württemberg | LUBW. State Agency for the Environment Baden-Württemberg – Hydrographic Service, Karlsruhe, Germany. https://udo.lubw.baden-wuerttemberg.de/public/ (last access: 12 Dec 2023). |
| Germany, Bayern | GKD. Bavarian State Office for the Environment – Hydrographic Service, Munich, Germany https://www.gkd.bayern.de/en/rivers/discharge/tables (last access: 12 Dec 2023). |
| Germany, Berlin | WB. Das Wasserportal Berlin: https://wasserportal.berlin.de/start.php (last access: 12 Dec 2023). |
| Germany, Hessia | HLNUG. Hessisches Landesamt für Naturschutz, Umwelt und Geologie. https://www.hlnug.de/ (last access: 12 Dec 2023). |
| Germany, Lower Saxony | NLWKN. Niedersachsischer Landesbetrieb fur Wasserwirtschaft, Kusten- und Naturschutz, http://www.wasserdaten.niedersachsen.de/ (last access: 12 Dec 2023). |
| Germany, North Rhine-Westphalia | ELWAS-WEB. Ministerium fur Umwelt, Naturschutz und Verkehr des Landes Nordrhein-Westfalen, https://www.elwasweb.nrw.de/elwas-web/ (last access: 12 Dec 2023). |
| Germany, Rhineland-Palatinate | MKUEM. Ministerum für klimaschutz, umwelt, energie und mobilität: Rheinland-Pfalz, Germany. https://wasserportal.rlp-umwelt.de (data received: 13 Mar 2023). |
| Germany, Schleswig-Holstein | Umweltportal. Schleswig-Holstein, Germany. https://umweltportal.schleswig-holstein.de/ (last access: 12 Dec 2023). |





| | |
|---|---|
| Germany, Saxony | ASOEAG. Saxon State Office for Environment, Agriculture and Geology: Datenportal fur Umweltdaten Sachsen (iDA), https://www.umwelt.sachsen.de/ (last access: 12 Dec 2023). |
| Sachsen-Anhalt | LHW. Landesbetrieb fur Hochwasserschutz und Wasserwirtschaft Sachsen-Anhalt, https://gld.lhw-sachsen-anhalt.de/ (last access: 12 Dec 2023). |
| Germany, Thüringen | LUBN. Landesamt für Umwelt, Bergbau und Naturschutz. Hochwasser Nachrichten Zentrale: Freistaat Thüringen. https://hnz.thueringen.de (data received: 13 Mar 2023). |
| Germany | BFG. Bundesanstalt für Gewässerkunde, Germany. https://www.bafg.de/DE/Home/homepage_node.html (data received: 13 Mar 2023). |
| Denmark | ODA. Overfladevandsdatabasen: Aarhus University, Denmark. https://odaforalle.au.dk/login.aspx (last access: 17 Jul 2023). |
| Estonia | GRDC. Global Runoff Data Center: River discharge data. Federal Institute of Hydrology, 56068 Koblenz, Germany. https://www.bafg.de/GRDC (last access: 01 May 2024 |
| Spain | CEDEX. Centro de Estudios y Experimentación de Obras Publicas: Anuario de aforos 2019-2020, Spain. https://ceh.cedex.es/anuarioaforos/demarcaciones.asp (last access: 12 Apr 2023). |
| Finland | FEI. Finish Environmental Institute, Finland. https://wwwp2.ymparisto.fi/scripts/kirjaudu.asp (last access: 10 Jul 2023). |
| France | BanqueHydro. Hydro Portail, France. https://www.hydro.eaufrance.fr/ (last access: 06 Jul 2023). |
| Great Britain | NRFA. National River Flow Archive API, United Kingdom. https://nrfaapps.ceh.ac.uk/nrfa/nrfa-api.html (last access: 07 Jul 2023). |
| Croatia | DHZ. Croatian Meteorological and Hydrological Service. https://hidro.dhz.hr/ (last access: 08 Oct 2023). |
| Hungary | "OVF. General Directorate of Water Management. https://ovf.hu/kozerdeku/adatigenyles (data received: 18 Aug 2023). |
| GRDC. | Global Runoff Data Center: River discharge data. Federal Institute of Hydrology; 56068 Koblenz Germany. https://www.bafg.de/GRDC (last access: 01 May 2024)" |
| Ireland | "EPA. Environmental Protection Agency; Ireland. https://epawebapp.epa.ie/hydronet/#Flow (last access: 27 Jun 2023). |
| OPW. Office of Public Works | Ireland. https://waterlevel.ie/hydro-data (last access: 27 Jun 2023)." |
| Iceland | Helgason, H. B. & Nijssen, B. LamaH-Ice: LArge-SaMple DAta for Hydrology and Environmental Sciences 574for Iceland, CUAHSI HydroShare (last access: 01 May 2024). 2023 |
| Italy, Emilia-Romagna | ARPAE Emilia-Romagna. Agenzia Prevenzione Ambiente Energia - Emilia-Romagna, Italy. https://simc.arpae.it/dext3r/ (last access: 04 Nov 2023). |



| Italy, ISPRA | ISPRA. Institute Superiore per la Protezione e la Ricerca Ambientale, Italy. http://www.hiscentral.isprambiente.gov.it/hiscentral/hydromap.aspx?map=obsclient, (last access: 30 December 2023). |
|---|---|
| Italy, Lombardia | ARPA Lombardia. Agenzia Regionale per la Protezione dell Ambiente - Lombardia, Italy. (data received: 17 Jun 2023). |
| Italy, Liguria | ARPAL Liguria. Agenzia Regionale per la Protezione dell Ambiente - Liguria, Italy. https://www.arpal.liguria.it (data received: 08 Jun 2023). |
| Italy, Sardegna | ARPA Sardegna. Agenzia Regionale per la Protezione dell Ambiente - Sardegna, Italy. https://www.sardegnaambiente.it/index.php?xsl=611&s=21&v=9&c=93749&na=1&n=10 (last access: 30 December 2023). |
| Italy, Toscana | ARPA Toscana. Agenzia Regionale per la Protezione dell Ambiente - Toscana, Italy. http://www.sir.toscana.it/consistenza-rete (last access: 16 Jun 2023). |
| Italy, Umbria | ARPA Umbria. Agenzia Regionale per la Protezione dell Ambiente - Umbria, Italy. https://annali.regione.umbria.it (last access: 22 May 2023). |
| Italy, Veneto | ARPAV Veneto. Agenzia Regionale per la Prevenzione e Protezione Ambientale del Veneto, Italy. https://www.arpa.veneto.it/ (data received: 30 Jun 2023). |
| Lithuania | GRDC. Global Runoff Data Center: River discharge data. Federal Institute of Hydrology, 56068 Koblenz, Germany. https://www.bafg.de/GRDC (last access: 01 May 2024 |
| Latvia | GRDC. Global Runoff Data Center: River discharge data. Federal Institute of Hydrology, 56068 Koblenz, Germany. https://www.bafg.de/GRDC (last access: 01 May 2024 |
| Moldova | GRDC. Global Runoff Data Center: River discharge data. Federal Institute of Hydrology, 56068 Koblenz, Germany. https://www.bafg.de/GRDC (last access: 01 May 2024 |
| N. Ireland | NRFA. National River Flow Archive API, United Kingdom. https://nrfaapps.ceh.ac.uk/nrfa/nrfa-api.html (last access: 07 Jul 2023). |
| Netherlands | RWS. Rijkswaterstaat waterinfo, The Netherlands. https://waterinfo.rws.nl/#/publiek/waterafvoer (last access: 07 Dec 2023). |
| Norway | NVE. Norwegian Water Resources and Energy Directorate, Norway. https://seriekart.nve.no (last access: 10 Jul 2023). |
| Poland | IMGW-PIB. Institute of Meteorology and Water Management - National Research Institute, Warszawa, Poland. https://danepubliczne.imgw.pl/introduction (last access: 30 Dec 2023). |
| Portugal | SNIRH. Sistema Nacional de Informação de Recursos Hídricos: Dados de Base, Portugal. 772https://snirh.apambiente.pt/index.php?idMain=2&idItem=1 (last access: 01 May 2024 |



| Romania | GRDC. Global Runoff Data Center: River discharge data. Federal Institute of Hydrology, 56068 Koblenz, Germany. https://www.bafg.de/GRDC (last access: 01 May 2024 |
|---|---|
| Serbia | GRDC. Global Runoff Data Center: River discharge data. Federal Institute of Hydrology, 56068 Koblenz, Germany. https://www.bafg.de/GRDC (last access: 01 May 2024 |
| Russia | GRDC. Global Runoff Data Center: River discharge data. Federal Institute of Hydrology, 56068 Koblenz, Germany. https://www.bafg.de/GRDC (last access: 01 May 2024 |
| Sweden | SMHI. Swedish Meteorological and Hydrological Institute, Sweden. https://www.smhi.se/data/hydrologi/ladda-ner-hydrologiska-observationer#param=waterdischargeDaily; stations=core (last access: 30 Dec 2023). |
| Slovenia | ARSO. Agencija Republike Slovenije za Okolje, Ljubljana, Slovenia. https://vode.arso.gov.si/hidarhiv/ (last access: 23 Jun 2023). |
| Slovakia | GRDC. Global Runoff Data Center: River discharge data. Federal Institute of Hydrology, 56068 Koblenz, Germany. https://www.bafg.de/GRDC (last access: 01 May 2024 |
| Ukraine | GRDC. Global Runoff Data Center: River discharge data. Federal Institute of Hydrology, 56068 Koblenz, Germany. https://www.bafg.de/GRDC (last access: 01 May 2024 |



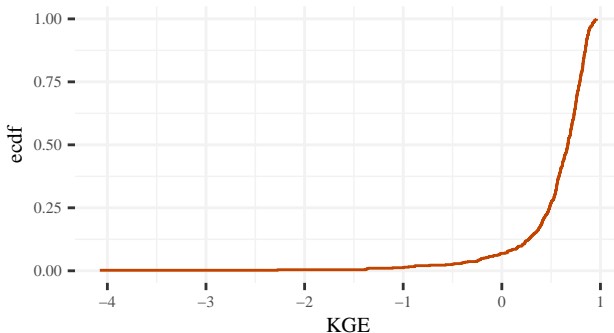

**Figure C1.** Empirical cumulative density function of the Kling–Gupta efficiency (KGE) for the test evaluation. Each point on the line represents the model performance for one of the 500 test basins.

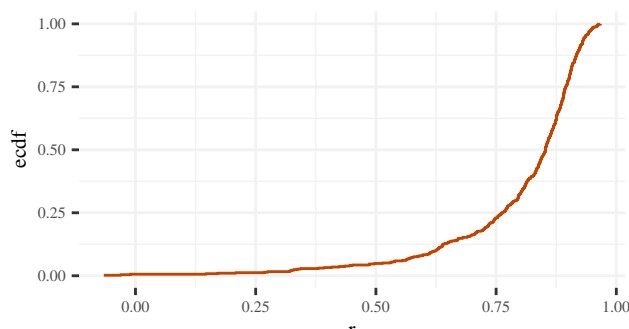

**Figure C2.** Empirical cumulative density function of Pearson's correlation coefficient between simulations and observation (r) for the test evaluation. Each point on the line represents the model performance for one of the 500 test basins.

## Appendix C: Other metrics for evaluation

This appendix shows the results for different evaluation metrics for the evaluation experiment presented in Sect. 2.3.3. Specifically, Fig. C1 plots the Kling–Gupta efficiency, Fig. C2 Pearson's correlation coefficient, and C3 the root mean squared error.

## Appendix D: Estimation of trend statistics

Our procedure for estimating the long-term trends statistics for yearly flood timing and yearly peak flow trends (Sect. 2.3.4) follows Blöschl et al. (2017) and Blöschl et al. (2019) respectively. This section partially mirrors the supplementary material Blöschl et al. (2017) and describes the technical details of our implementation.



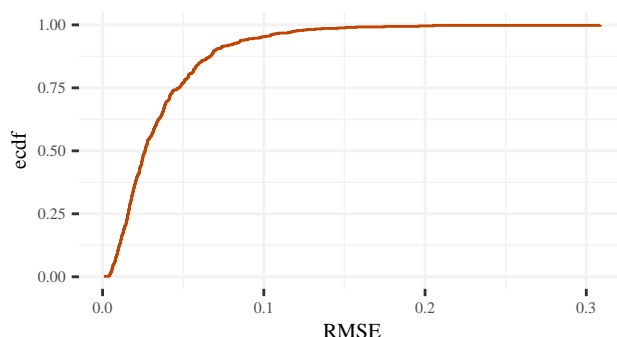

**Figure C3.** Empirical cumulative density function of the root mean squared error (RMSE) for the test evaluation. Each point on the line represents the model performance for one of the 500 test basins.

### D1   Yearly peak-flow trends

For the flood trend analysis part we follow Blöschl et al. (2017): First, we extract a series of observerations that consists of the highest peak discharge recorded in each calendar year (i.e., the annual maximum peak flow) from EARLS. Then, we estimate
the trend in each series using a robust approach and interpolate the trend estimates using Kriging. Blöschl et al. (2017) only provide the data for their experiments, while Blöschl et al. (2019) provide data and R-code for their experiments. Hence, for this demonstration we modified their code to stay as close as possible to their results. The robust estimation is achieved by using the Theil–Sen slope $\beta$ (Theil, 1950; Sen, 1968):

$$\beta = \text{median}\left(\left\{\frac{Q_i - Q_j}{i - j}\,\middle|\, i, j \in \mathcal{J}\,\text{and}\, i \neq j\right\}\right),$$  (D1)

where $Q_z$ indicates the maximal streamflow of a given year $z$ and $\mathcal{J}$ contains the indices of all the years from 1960 to 2010 — which corresponds to the time span that Blöschl et al. (2019) use.

### D2   Flood timing analysis and trends

The demonstration of the flood timing analysis follows Blöschl et al. (2017). Specifically, we reproduce two of their investigations: (1) an examination of long-term trend of the flood-timing and an analysis of the average flood timing over Europe
(represented by Fig. 1 and Fig. 3 in Blöschl et al. (2017); and Fig. 7 and Fig. 8 in our contribution). Following their procedure, we first compute for each station the average day $D$ within a year where peak flows have occurred during the observation period. And, to account for the cyclic nature of yearly data all calculations are performed using circular procedures/statistics. As Blöschl et al. (2017) we only take the stations for which the null hypothesis of circular uniformity (which is assessed with Kuiper's test) is rejected with a significance level of $\alpha = 0.1$. For this contribution no code is available. Hence, we use the
description in the supplementary material to modify the code from Blöschl et al. (2019) for the trend examination. This means,



that code for the average flood timing analysis is largely from ground up (according to the provided documentation from the supplementary material).

We convert date of occurrence $D_i$ of a flood in year $i$ into an angular value using:

$$\theta = \frac{2\pi}{m_i} \quad \text{with} \quad 0 \leq \theta_i \leq 2\pi, \tag{D2}$$

where $m_i$ is the number of days for year $i$, and $D_i$ gives the corresponding day of the year so that $D_i = 1$ corresponds to January 1 and $D_i = m_i$ to December 31.

We compute the average date of occurrence $D$ of a flood at a station as:

$$\bar{D} = \begin{cases} \tan^{-1}\left(\frac{\bar{y}}{\bar{x}}\right)\frac{\bar{m}}{365} & \text{for} \quad \bar{x} > 0, \bar{y} \geq 0, \\ \left(\tan^{-1}\left(\frac{\bar{y}}{\bar{x}}\right) + \pi\right)\frac{\bar{m}}{2\pi} & \text{for} \quad \bar{x} \leq 0, \\ \left(\tan^{-1}\left(\frac{\bar{y}}{\bar{x}}\right) + 2\pi\right)\frac{\bar{m}}{2\pi} & \text{for} \quad \bar{x} > 0, \bar{y} < 0. \end{cases} \tag{D3}$$

Here, the arc-tangens $\tan^{-1}$ yields the angle in radians, $x$ and $y$ are the cosine and sine components of the average date, $m$

is the average number of days per year (which we re-calculate to account the fact that some years had no data), and $n$ is the total number of flood peaks at that station. That is:

$$\bar{x} = \frac{1}{n}\sum_{i=1}^{n}\cos(\theta_i),$$

$$\bar{y} = \frac{1}{n}\sum_{i=1}^{n}\sin(\theta_i),$$

and

$$\bar{m} = \frac{1}{n}\sum_{i=1}^{n}m_i.$$

Lastly, Blöschl et al. (2017) define concentration $R$ of the date of occurrence around the average date as:

$$R = \sqrt{\bar{x}^2 + \bar{y}^2} \quad \text{with} \quad 0 \leq R \leq 1. \tag{D4}$$

Indeed, the mapping of $\bar{D}$ and $R$ constitutes our reproduction of the average flood timing analysis from Blöschl et al. (2017).

**D3  Trends in timing**

For the timing trend estimation we use the same adjusted Theil-Sen slope estimator as reported by Blöschl et al. (2017). The computation is similar to the one given by Eq. (D1), but adds a correction factor $k$ to account for the circularity of the task:

$$\beta_{\circ} = \text{median}\left(\left\{\left.\left|\frac{Q_i - Q_j + k}{i - j}\right| \, \right| \, i, j \in \mathcal{J} \text{ and } i \neq j\right\}\right) \quad \text{with} \quad k = \begin{cases} -\bar{m} & \text{if} \quad D_j - D_i > \bar{m}/2, \\ \bar{m} & \text{if} \quad D_j - D_i < -\bar{m}/2, \\ 0 & \text{otherwise.} \end{cases} \tag{D5}$$





Here, the Theil-Sen slope $\beta_\circ$ has units of days per year. Thus to get an estimate for the 10 year period reported in Blöschl
et al. (2017) it has to be scaled accordingly. Lastly, we used the same Kriging approach as we do for the yearly peak-flow
trends to get a map of the large-scale spatial patterns within Europe.

*Author contributions.* JZ and DK developed the idea, conceptualization, and method of the paper. DK did all LSTM simulations. PM conducted the mHM simulations. TN and FF helped with the EStreams setup and model realizations. MG provided the setup for the ungauged basins as well as additional model simulations, control experiments, and checks. All authors were involved in the writing of the paper.

*Competing interests.* The contact author has declared that none of the authors has any competing interests.

*Acknowledgements.* Daniel Klotz and Jakob Zscheischler acknowledge funding from the Helmholtz Initiative and Networking Fund (Young Investigator Group COMPOUNDX, grant agreement no. VH-NG-1537). We thank Frederik Kratzert for his input with the data and the modeling setup, as well as Rohini Kumar for his help with the mHm model, and Emanuele Bevacqua for discussing intricacies of the E-OBS data quality with us. We acknowledge the E-OBS dataset and the data providers in the ECAD project (https://www.ecad.eu).





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
