# Peer review of "EARLS: A runoff reconstruction dataset for Europe"

_Earth System Science Data, 2024_

## Referee Comment (RC1)

**Review of "EARLS: A runoff reconstruction dataset for Europe"**

This paper presents an interesting contribution called EARLS. EARLS provides daily streamflow reconstructions (with uncertainty estimates) for over 10,000 European basins from 1953 to 2023, using a single LSTM-based rainfall–runoff model trained on data from more than 5,000 basins. The quality of the dataset is assessed through comparisons with held-out sets of basins and a qualitative evaluation of peak flows and flood timing between EARLS and results previously reported by Blöschl et al. EARLS is publicly available and serves as good example of how Deep Learning can be used to generate large-sample "datasets" with improved spatial and temporal coverage compared to existing observational datasets (e.g., EStreams in the case of Europe). The paper is well-written, and the analyses appear robust; thus, EARLS has the potential to be a valuable contribution to the hydrological community. In principle, I support the publication of this work without many changes. However, I also provide some related comments below, which I encourage the authors to consider to further strengthen the work and better justify its publication in ESSD.

**Where does data end and model estimates start?**

I agree with the opening paragraph that data are central to hydrological science. Observations are the primary reference for enabling hypothesis testing and advancing science (e.g., through studies utilizing LSTM models). However, the distinction between where "data" end and "model estimates" begin remains ambiguous. While it is true that observed hydrographs are not entirely free from assumptions (e.g., the construction of a rating curve), considering (LSTM) reconstructions as data on the same level is a significant leap. Are "reconstructions" (at least the parts where they provide estimates of data) not inherently limited in this regard? They offer gap-filled estimates dictated by the model, but they lack the inclusion of scientific testing that could uncover explicit new process knowledge beyond what is already encoded in the model.

The availability of streamflow estimates provided by EARLS is undoubtedly convenient, but is convenience what hydrological science needs? Scientific ideas should be tested using observations, and existing datasets (e.g., EStreams) provide such data. Gap-filled analyses should only be used when original data are insufficient, and even then, they are only useful if there is strong evidence that the gap-filling process accurately captures the specific hydrological behavior being tested. Consequently, a generic statement such as "Our results suggest that EARLS as a dataset is well-suited for large-sample hydrological studies in Europe" seems misplaced, as its suitability ultimately depends on the specific goals of the hydrological study.

Right now, these issue remains largely undiscussed, except for some comments in the conclusions. Is this the right place to briefly bring this up or should this be done earlier and more extensively?

**How human-affected are catchments with <4 dams?**

EStreams provides runoff estimates for catchments that have been screened for human impacts (fewer than four dams). However, in reality, there are over a million obstructions in European rivers (see: https://amber.international/european-barrier-atlas/). While not all of these obstructions significantly affect streamflow, the cutoff of four dams—regardless of catchment size—seems rather ad hoc. Why isnt a metric like dam density (e.g. per unit catchment area) or a similar approach used instead?

**Comparisons are with the interesting studies of Bloschl et al. but are these the most informative comparisons?**

Seasonal flooding is largely driven by climate conditions and has minimal connection to local landscape factors (e.g., see the geographical clusters in Fig. 7 and earlier studies attributing flood drivers). Could you devise a more challenging test that better demonstrates the utility of EARLS? For instance, you could quantify differences between observed and EARLS behaviors across several distinct and complementary hydrological signatures, ideally including a signature that is less spatially autocorrelated and that primarily reflects local differences between catchments rather than large-scale climatic gradients. While I understand the Bertola metric is used, this result is not presented spatially and also focuses on flooding. For example, I do not know whether EARLS is currently suitable for low-flow studies (or any other aspect of flow than annual high flows)

**Is Fig 9 a reason to celebrate success, or does it highlight strong limitations?**

I agree that some first-order inferences from Blöschl et al. are also apparent in EARLS. However, significant differences, including at larger scales, are evident (for example, many regions show opposing trends—consider Scandinavia—and I estimate that roughly 30% of the map displays inconsistent, opposing trends).

You state, you *"[...] argue that the results of our qualitative assessment show the merit our* *the*\* *EARLS data for scientific inquiry and corroborate the quality of the simulations.."* However, to me, Fig. 9 suggests that I would be extremely hesitant to rely on EARLS for such analyses (given the inconsistencies highlighted above). Am I being too pessimistic?

\*remove "the"

**Why is the comparison in section 3.4/fig9 kept qualitative?**

The comparison is currently qualitative, but transitioning to a quantitative analysis would be straightforward. Differences could be quantified and mapped at the pixel scale, while values across a continental scale could be compared using metrics such as mean absolute error, $R^2$, or other statistical measures. At present, the approach feels unnecessarily qualitative.

**Please check lines 293-305.**

(Maybe I found it challenging to follow your reasoning because I encountered this part towards the end of the review.) However, this section appears to be imprecisely formulated, leaving the reader too often to guess the intended meaning behind the words.

**Data screening**

I understand that the Estreams data has already undergone screening before publication, and you apply additional criteria.

However, it's important to note that some (artefactual) patterns likely persist in the data you're considering.

For instance, in the case of ES000454, it appears that patterns changed in 1966, coinciding with the construction of a dam. Additionally, the data for this station shows repeated nonzero values between 1973 and 1982.

I understand that data screening can be very time consuming, but did you also visually inspect the remaining hydrographs? This would be feasible for 5000 stations.

**Detailed comments**

- Fix the type on the first line of Eq. D3. (*m/2pi and not *m/365?)
- I understand that this paper is not about testing the trends in flood seasonality across Europe, but please note the methodological obscurity of Eq. D5: It is clear that one needs to take into account the circularity of the calendar year, and therefore, it seems logical that the variable *k* is introduced. However, for a trend in time, two processes can be more than half a year apart when considering their difference (and thus trend). Consider an example where flood peaks on average occur July 1st, but in one year the flood occurred early in the year (e.g. 15 March) and the subsequent year it was a late occurrence (e.g., 15 October). This suggests, for this pair, a trend towards later flooding, but the correction by *k* would qualify this a tendency towards earlier flooding.  This could be solved by using a wrapping function, but applying this is probably outside the scope of the manuscript as your goal is to test how it compares to the results of bloschl.

- L6: are **often** well suited to provide predictions in ungauged basin" (as sometimes they clearly are not).
- L6-8: consider stating what the comparison shows, rather than that the comparison is made (mention the result/conclusion, not only the method)
- L90: out of curiosity: elevation is used, but elevation itself is not affecting hydrology, it are physical properties of the catchment that will correlate with elevation that do this. Do you have any idea which physical factors these are? (No changes required, just curious).
- L173: since NSE cannot really be compared between places I would not talk about "best and worst"but about "highest and lowest".
- L173: These values are not randomly distributed (Fig. 4): specify you talk about a distribution in space (or geographically)
- Caption Figure 8: "but show more detail." I understand they show more detail, but given the performance of the model I do not know if these "details" are right or artefacts, so I would be careful to emphasize this.
- L294: what do you really mean by: virtual variable?
- L296: what do you really mean by: is a new qualitative dimension?
- L296 "The job of the model is here to extract the information that the meteorological signals contain about the streamflow and act as a virtual sensor" is to me a vague statement.
- L297-299: "This is certainly not a replacement for real data — on the contrary, our ability to enable models in this way heavily depends on the availability of large amounts of diverse and highly qualitative data (Kratzert et al., 2024)." I do not fully follow you here.

---

## Referee Comment (RC2)

**Review**

Klotz, D., Miersch, P., do Nascimento, T. V. M., Fenicia, F., Gauch, M., and Zscheischler, J.: EARLS: A runoff reconstruction dataset for Europe, Earth Syst. Sci. Data Discuss. [preprint], https://doi.org/10.5194/essd-2024-450, in review, 2025.

Dear Daniel and others,

It was a pleasure to review your manuscript on *"EARLS: A runoff reconstruction dataset for Europe"*, submitted to the Earth System Science Data journal. I found your study to be well-structured, informative, and a very valuable contribution to the field.

I have provided a list of minor comments for your consideration. Most of these do not require urgent revisions, but I have highlighted in blue those that may benefit from some additional attention.

My main comments focus on the following aspects:
- Slight inconsistencies between datasets – While these are not major issues and likely do not require changes, I would appreciate your expert opinion on what potential impacts these inconsistencies might have on your findings.
- Figure clarity – Some suggestions are provided to improve comparisons and ease interpretation.
- Simulation data and uncertainties – Comments regarding the data shared to ensure transparency and clarity.

Overall, I am recommending minor revisions. I appreciate the effort you have put into this work and would be happy to have another look at the revised version.

Best regards,
Julie Mai

Detailed comments:

Section 1:
- None

Section 2:
- Line 74: "not able to produce a simulation for 227" —> Why? Are these 227 part of the 14,161 basins without data gaps, or the 2,655 with gaps, or additional ones?
- Line 83: "The basin shapes are either derived from EStreams (do Nascimento et al., 2024) or HydroATLAS (Linke et al., 2019)" —> For a given basin, what made you decide which one to use? If you specify this later please refer to that section here.
- Line 84: "We use 4 dynamic inputs" —> You state later that EOBS has more forcing variables. Are the other ones not contributing to a better performance? Did you test other variables? I think it would be great to have some additional information on this here.
- Line 88: "[EOBS is] spanning 1 January 1950 to present." —> Your dataset starts in 1953 though. Why are the first 3 years skipped. I think that's fine. It would just be good to have some information why you skipped 3 years.
- Line 90: "we use 13 attributes that we aggregate from HydroATLAS, namely: basin area, the average elevation, the average slopes, the average stream gradient, the average long-term air temperature, the minimum long-term air temperature, the maximum long- term air temperature, a global aridity index" —> I am wondering what happens if you are using a basin shape from EStreams (not HydroATLAS) to aggregate forcing etc but then you use basin characteristics that are (technically) for a (potentially) different basin shape. For example, basin area may be different but also meteorologically based basin characteristics would not be consistent. Further, why are the meteorologically based attributes to directly derived from the forcing dataset? I'm sure this is all just minor differences but some kind of note for the reader may be helpful.
- Figure 2: The positioning of the three-parameter Laplace distribution of simulation results just at the end of the LSTM time line looks bit like there is only one set of the three parameters estimated. It maybe clearer if each of the orange LSTM "boxes" would have an arrow (or connection) to the gamma, b, and tau. Or maybe place an index "t" at each of the three parameters to show they are time dependent?
- Line 146: "seperately delineated gauged basins" —> What does that mean? Basins have another shape than you used for EARLS? Doesn't that mean that the forcings are (slightly) different? What impact may it have that EARLS estimates are for a potentially slightly larger or smaller basin than mHM given that it is another basin shape? Also, typo: "seperately" —> "separately"
- Table 1: You list the datasets used to setup mHM. Technically these could be used to derive the static basin attributes for the LSTM. How different are estimates from those datasets compared to what you use from HydroATLAS? Would these differences matter?

Section 3:
- "potentially difficult to predict ungauged basins" —> It is stated that all basins have "natural" streamflow (line 53). I was hence expecting that these are mostly pristine basins with low human impact. Is that correct? What would make these basins then "potentially difficult to predict"? It would be great if some reasons why a basin may be "potentially difficult to predict" would be added to the manuscript. Also, out of curiosity, how do you know that basins have "natural" streamflow. Is that a HydroATLAS attribute for basins? If so, maybe mention that?
- Line 169: I think it would be helpful for the reader if you could start the section with repeating the time period you evaluate here.
- Line 171: suggestion: "opposed to Kratzert et al., 2021" —> "opposed to Kratzert et al., 2021 using multiple facing datasets"
- Line 172: "These results are similar to the ones in Kratzert et al. (2019a), but worse than the ones from Mai et al. (2022)." —> Would it be possible to state how many where below 0.0 in those two studies? It would help for comparison.
- Figure 3: The validation data line is barely visible. Maybe use thinner lines for all three and remove (some) gridlines? It would also be nice to state clearly how many basins are in each of the three sets. Maybe add to the legend (e.g., "test data (N=500)")?
- Figure 4:
  - The caption states that "The colors mirror Fig. 3". But figure 3 has colours for test, training and validation data. Figure 4 shows NSE performances of the validation (?) data.
  - I think figure 3 and 4 should be combined since they are somewhat connected. The continuous colourbar in figure 4 is not really needed I think. I can only distinguish 3 colours- blue, red and purple. I would just pick 3-4 distinct colours (or the ones you use in for example Figure 8) and then use those to get the point across. Also, I think the colourbar needs to be open ended at the lower end as I am assuming that there is values smaller than -0.5 that are also coloured in red?!
  - It would be great to exactly state which basins are in figure 4. I am guessing validation basins. I would use the same exact wording as used in figure 3.
- Line 175-190: I really like this discussion. It states a lot about the methodology (data pre-processing and filtering of basins) that I was expecting beforehand (see comments above). I think it maybe advisable to have a separate section called "data pre-processing" where all this is placed before we dive into model performance etc.
- Figure 5: caption suggestion:
  - "Bertola distances" —> "Bertola distances d"
  - Please add to the caption what panel a and b depicts.
  - Typo: "D_{<0.8}" —> "D_{>0.8}" (in caption)
- Line 199: "Tn" —> "In"
- Line 201: "roughly starting at the $60^{th}$ percentile" —> I would actually say much earlier; like around $37.5^{th}$ percentile. Unless the authors look at another feature or criterion to determine when models start looking similar in performance… It maybe helpful to have those lines added as

horizontal lines in Figure 6 for the reader to better be directed towards what they are supposed to see.

- Line 201: "For the remaining 15%" —> I am not sure which remaining 15%.
  - 0-15th = EARLS and default are similar
  - 60-100th (I suggest 35-100th)= EARLS and local are similar
  - 15-60th (I suggest 15-35th) = in between -> but this is 45% (unless the authors agree that it should be around 20% remaining)
- Figure 7: I love that figure!
- Line 207: "We encourage readers to compare our version with these depictions" —> Please be aware that the publication is not open access. So, it may limit the ability of readers to actually do this. :( I am assuming it is not reasonable to recreate the figure with the Blöschl data and have them for comparison in the manuscript?
- Figure 8: The caption is not consistent with the colorbar. "Positive trends are depicted in red" but they are blue and vice versa. Unless the legend shows Blöschl minus EARLS estimates and then you talk about EARLS minus Blöschl in the figure caption. In any case it's confusing.
- Line 219: "the original version shows slightly positive trends" —> Isn't figure 9a all "red" in Scandinavia which is negative values which means negative trends (see caption figure 9). There is some sort of mix-up what positive and negative means I think. There maybe more in this paragraph but I leave it to the authors to revise them without pointing each one out here.
- Line 219: "positive trends" and "negative trends" —> in general, I am not sure if I would refer to them as "positive" and "negative". I am understanding that positive values indicate more floods and I am not sure if that's "positive". Maybe refer to them as "trends of increasing number of floods" or "increasing trend of floods" or something...
- Line 224: "Some differences can be explained by data availability" —> Do you think it may be helpful to actually plot the gauge stations used for the two datasets in figures 9a and 9b? It may underline your point that differences appear where more data are available while the other dataset lacks observations.
- Figure 9: Would it be possible to use the same colorbar as Figure 8?
- Figure A1 and A2: I highly recommend to merge these two figures into one figure with two panels. This would make it much easier to compare the workflows; especially when the box-diagrams (which are beautiful) are arranged in the same way. I think the "selection" step in either figure needs to be part of the methodology. Currently this is a bit vague and distributed. It's such a curial step that it should be easily findable. I later found some of that in the Appendix but maybe move it to the methods or at least refer to this section of the appendix early on in the methods. Also, there should be a comparable section like this for the section of "engaged" basins, right?
- Table B1: Wow!
- Figure C1 to C3: I think these three can be merged into one figure with three panels. It would be easier to compare them if they are next to each other. Also, the KGE could just run from -2 or -1 to 1. This way one would see more of the actual interesting part of the rising limb.

Section 4:
- Line 256: You may want to link to DOI "10.5281/zenodo.13864842" which would always point to the latest version of the dataset.
- Line 263: The constructions folder is stated to include CSV files with at least date and simulation in mm/day. That is great. I am however wondering if it would be possible to include some information about the uncertainty estimates. I think these are a major selling point of this dataset and it is stated that it is included later (line 289-291: "each time step EARLS provides a conditional uncertainty estimate — which can, for example, be used to compute the likelihood of a given model" (which I really like). I was however not able to download the full 33GB dataset and check if there may be a file that contains the uncertainty information. The estimated download time was 26 hours which seemed too much to wait…
  - If the uncertainty data are contained, please make more clear where one would find these data.
  - If it's not included maybe make more clear that a user would need to setup an LSTM themselves and train it and then get those estimates themselves.
- Is the download always taking so long or is it just me? An idea would be to have a mini-example with 3-5 basins in a separate (much smaller) zip such that people could download that to see if it contains what they would expect, and setup workflows while they wait for the entire pack to download?! Up to the authors, of course.

Section 5:
- Line 288: "11 thousand" —> "11,000"

Acknowledgements:
- Line 433: "mHm" —> "mHM"

---

## Author Comment (AC1)

**Reviewer: Wouter Berghuijs**

This paper presents an interesting contribution called EARLS. EARLS provides daily streamflow reconstructions (with uncertainty estimates) for over 10,000 European basins from 1953 to 2023, using a single LSTM-based rainfall–runoff model trained on data from more than 5,000 basins. The quality of the dataset is assessed through comparisons with held-out sets of basins and a qualitative evaluation of peak flows and flood timing between EARLS and results previously reported by Blöschl et al. EARLS is publicly available and serves as a good example of how Deep Learning can be used to generate large-sample "datasets" with improved spatial and temporal coverage compared to existing observational datasets (e.g., EStreams in the case of Europe). The paper is well-written, and the analyses appear robust; thus, EARLS has the potential to be a valuable contribution to the hydrological community. In principle, I support the publication of this work without many changes. However, I also provide some related comments below, which I encourage the authors to consider to further strengthen the work and better justify its publication in ESSD.

Dear Wouter,

thank you for your thoughtful review. The review provided a lot of thoughts related to the big picture of the dataset itself and we highly appreciate this kind of feedback — even if it does not always lead to direct changes in the manuscript, we believe it improved the overall argument.

**Where does data end and model estimates start?**
I agree with the opening paragraph that data are central to hydrological science. Observations are the primary reference for enabling hypothesis testing and advancing science (e.g., through studies utilizing LSTM models). However, the distinction between where "data" end and "model estimates" begin remains ambiguous. While it is true that observed hydrographs are not entirely free from assumptions (e.g., the construction of a rating curve), considering (LSTM) reconstructions as data on the same level is a significant leap. Are "reconstructions"

(at least the parts where they provide estimates of data) not inherently limited in this regard? They offer gap-filled estimates dictated by the model, but they lack the inclusion of scientific testing that could uncover explicit new process knowledge beyond what is already encoded in the model.

We mostly agree with this part of the feedback and find the current discussion of our manuscript reflects it. For example, the statement that "streamflow measurements are of higher quality than simulations, despite being constructed too" is literally an argument that we make. We absolutely do not claim that our reconstructions are at the same level of accuracy as observations and will make sure to revise the discussion so that this becomes crystal clear to all readers.

There are, however, two small aspects that we would like to clarify: Firstly, data does not end where model estimates begin. Estimates are, by definition, data. Secondly, the property that one can only extract the information that is contained in the data is a fundamental property of information in data (see for example Chpt. 2.8 in Cover and Thomas, 2006). As such, it is shared by all forms of data irrespective of their provenance and independently how much scientists try to analyse or process the data.

The availability of streamflow estimates provided by EARLS is undoubtedly convenient, but is convenience what hydrological science needs? Scientific ideas should be tested using observations, and existing datasets (e.g., EStreams) provide such data. Gap-filled analyses should only be used when original data are insufficient, and even then, they are only useful if there is strong evidence that the gap-filling process accurately captures the specific hydrological behavior being tested.

We agree with this reflection in spirit. Observations and experiments are the ultimate arbiter in science. At the same time science is much more than a mere collection of ideas, factoids, and data points of observables. It is conjecture; it is exploration; it is the weaving together of ideas, understanding, and empirical reality; it is the generation, management, and updating of knowledge; it is much more — and in all of this in-silico results can (and, as a matter of fact did already!) help.

Hence, we would (a) like to avoid dictating to scientists what they require or not, and (b) argue that the argumentation is formulated too broadly. Many sciences already make extensive use of large-scale simulation datasets. Perhaps the most famous example from earth science is climate science (but there are many more). We are not aware of any distinguishing factor that would hinder hydrologists to also account for evidence from simulations—keeping in mind that the resulting insights stem from simulations, as discussed in the answer above (and, as a matter of fact, in practice they do).

Consequently, a generic statement such as "Our results suggest that EARLS as a dataset is well-suited for large-sample hydrological studies in Europe" seems misplaced, as its suitability ultimately depends on the specific goals of the hydrological study. Right now, these issue remains largely undiscussed, except for some comments in the conclusions. Is this the right place to briefly bring this up or should this be done earlier and more extensively?

We appreciate the concern and recommendations. However, our manuscript extensively discusses these issues; particularly we would say that even more than any other comparable dataset published on ESSD. For instance, we show that EARLS can be used to investigate long-term streamflow trends in similar ways as done in previous studies. We therefore believe that a more extensive discussion would not improve the manuscript.

**How human-affected are catchments with <4 dams?**
EStreams provides runoff estimates for catchments that have been screened for human impacts (fewer than four dams). However, in reality, there are over a million obstructions in European rivers (see: https://amber.international/european-barrier-atlas/). While not all of these obstructions significantly affect streamflow, the cutoff of four dams—regardless of catchment size—seems rather ad hoc. Why isn't a metric like dam density (e.g. per unit catchment area) or a similar approach used instead?

We appreciate your concern, and indeed from a modelling perspective this is slightly arbitrary. We could also train our model on the whole dataset without pre-filtering

and derive predictions/evaluations. However, we decided to do some data curation because it is associated with better training behavior and performance of machine learning models. And, questions about how and how much one should curate the data are always a matter of convention and taste. There are infinitely many criteria to choose how to filter the data and our model does not strictly require perfectly unaffected streamflow. The one proposed here is one realization of these criteria.

That said, we like the proposed idea per se. As we write in our manuscript, EARLS is meant as a living dataset. Hence, we happily support other versions that use different delineation criterions for future expansions of the dataset. As such, we are happy to include such criteria first into a new EStreams version and then build a new EARLS version out of it. For the revised version of the manuscript we will mention this idea as future work.

**Comparisons are with the interesting studies of Bloschl et al. but are these the most informative comparisons?**

Seasonal flooding is largely driven by climate conditions and has minimal connection to local landscape factors (e.g., see the geographical clusters in Fig. 7 and earlier studies attributing flood drivers). Could you devise a more challenging test that better demonstrates the utility of EARLS?

We argue that these comparisons are suited to show the utility of EARLS. They certainly go well beyond all published hydrological datasets that we are aware of.

For instance, you could quantify differences between observed and EARLS behaviors across several distinct and complementary hydrological signatures, ideally including a signature that is less spatially autocorrelated and that primarily reflects local differences between catchments rather than large-scale climatic gradients. While I understand the Bertola metric is used, this result is not presented spatially and also focuses on flooding. For example, I do not know whether EARLS is currently suitable for low-flow studies (or any other aspect of flow than annual high flows)

It is not within the scope of our role as data creators to develop demonstrations for the wide array of potential hydrological applications. Given the diversity and breadth of possible uses for the dataset—many of which we may not yet be aware of—we

focus on ensuring the dataset is robust and well-documented, allowing researchers to tailor its application to their specific needs.

**Is Fig 9 a reason to celebrate success, or does it highlight strong limitations?**
I agree that some first-order inferences from Blöschl et al. are also apparent in EARLS. However, significant differences, including at larger scales, are evident (for example, many regions show opposing trends—consider Scandinavia—and I estimate that roughly 30% of the map displays inconsistent, opposing trends). You state, you "[...] argue that the results of our qualitative assessment show the merit our the* EARLS data for scientific inquiry and corroborate the quality of the Simulations.." However, to me, Fig. 9 suggests that I would be extremely hesitant to rely on EARLS for such analyses (given the inconsistencies highlighted above). Am I being too pessimistic?

We appreciate your concern, but we have the impression that this question arises from a misunderstanding of what the data from Blöschl et al. (2017, 2019) constitute. It is important to realize that there is no ground-truth to compare with in the first place. Consider the following: The maps from Blöschl et al. (2017) are constructed with the help of a small number of highly processed point sources, which are rasterized over large areas without observation by using a "gap filling" approach based on kriging. When we say gap filling, we use the terminology by the reviewer here, but we note that this filling is absolutely model based. Our reproductions with EARLS use the same gap filling mechanism, but the (naive) kriging is supported by a much denser distribution of many more support points. Since it was mentioned in the comment we take Scandinavia as an example (see Figure A1 below): If we consider a box estimate of the measurement station from Blöschl et al. (i.e., points that lie within a certain box of latitudes and longitudes) we get approximately 300 (here we rounded up in the decimals and neglect that they filter out specific stations for some of their analyses). In comparison, EARLS contains around 2500 simulated stations in the same area (here we rounded down in the hundreds). In that sense, 30% difference might not be bad. Either way, there is no ground truth to compare with. Since we can see where the reviewer came from we will add this discussion to the appendix of the revised manuscript.

[Figure]

(a) measurement stations from Blöschl et al. (2019)     (b) EARLS simulated stations

**Figure A1**. Comparison of the density of support points for the kriging interpolation exercise. Plot (a) shows the reference points from Bloesch et al. (2019), plot (b) the simulated station from EARLS. The colored areas in both (a) and (b) refer to the same subselection of the data based on boxing a given latitude and longitude.

*remove "the"

Thank you.

**Why is the comparison in section 3.4/fig9 kept qualitative?**

The comparison is currently qualitative, but transitioning to a quantitative analysis would be straightforward. Differences could be quantified and mapped at the pixel scale, while values across a continental scale could be compared using metrics such as mean absolute error, R2, or other statistical measures. At present, the approach feels unnecessarily qualitative.

This question probably goes back to the misunderstanding that caused the previous question. In short, this section is kept qualitative, since our goal is to show similarities in patterns. None of the proposed metrics capture that aspect. Not only that, using such metrics in this context would easily mislead readers into thinking that the Blöschl et al. (2017, 2019)  figures are a form of ground truth. They are not. In the revised manuscript we will make sure that this point is clear to the readers.

**Please check lines 293-305.**

(Maybe I found it challenging to follow your reasoning because I encountered this part towards the end of the review.) However, this section appears to be imprecisely formulated, leaving the reader too often to guess the intended meaning behind the words.

Thank you. We will go over it again.

**Data screening**

I understand that the Estreams data has already undergone screening before publication, and you apply additional criteria.

However, it's important to note that some (artefactual) patterns likely persist in the data you're considering.

For instance, in the case of ES000454, it appears that patterns changed in 1966, coinciding with the construction of a dam. Additionally, the data for this station shows repeated nonzero values between 1973 and 1982.

I understand that data screening can be very time consuming, but did you also visually inspect the remaining hydrographs? This would be feasible for 5000 stations.

We did visually inspect many hydrographs by randomly sampling different time-spans from different years.

Additionally, similarly to EARLS, EStreams is expected to be constantly updated, and we expect to have a new version with such visual inspection performed to all 17,000+ stations in the near future. As such, we will be happy to include such criteria first into a new EStreams version and then build a new EARLS version out of it—after the publication of the first version is finished. We will refer to this as future work in the revised manuscript.

**Detailed comments**

- Fix the type on the first line of Eq. D3. (*m/2pi and not *m/365?)

Thank you for pointing this out. We will correct this.

- I understand that this paper is not about testing the trends in flood seasonality across Europe, but please note the methodological obscurity of Eq.

> D5: It is clear that one needs to take into account the circularity of the calendar year, and therefore, it seems logical that the variable k is introduced. However, for a trend in time, two processes can be more than half a year apart when considering their difference (and thus trend). Consider an example where flood peaks on average occur July 1st, but in one year the flood occurred early in the year (e.g. 15 March) and the subsequent year it was a late occurrence (e.g., 15 October). This suggests, for this pair, a trend towards later flooding, but the correction by *k* would qualify this a tendency towards earlier flooding. This could be solved by using a wrapping function, but applying this is probably outside the scope of the manuscript as your goal is to test how it compares to the results of Blöschl.

Even if there are peculiarities in the approach we would like to keep them in our reproductions to make the comparison variable. We will, however, address this concern in the revised manuscript and mention that the computation is not necessarily intuitive.

As a side note: If we assume that the two events are the same, both the numerator and denominator would be negative. Hence, the tendency would be towards later flooding not earlier ones. Either way, as the reviewer points out himself, it has little to do with our comparison.

> - L6: are **often** well suited to provide predictions in ungauged basin" (as sometimes they clearly are not).

Ok.

> - L6-8: consider stating what the comparison shows, rather than that the comparison is made (mention the result/conclusion, not only the method)

We will revise this part as proposed.

> - L90: out of curiosity: elevation is used, but elevation itself is not affecting hydrology, it are physical properties of the catchment that will correlate with elevation that do this. Do you have any idea which physical factors these are? (No changes required, just curious).

No. Unfortunately, we do not (in general the factors do not just have to be physical, e.g., they can also be biological or anthropogenic).

- L173: since NSE cannot really be compared between places I would not talk

The goal here is to provide a ballpark number. We will make sure to emphasise that.

- about "best and worst"but about "highest and lowest".

Thank you. We will correct this.

- L173: These values are not randomly distributed (Fig. 4): specify you talk about a distribution in space (or geographically)

We will clarify by rephrasing to "These values are not randomly distributed in space (Fig. 4)"

- Caption Figure 8: "but show more detail. " I understand they show more detail, but given the performance of the model I do not know if these "details" are right or artefacts, so I would be careful to emphasize this.

The details are actually important in that they show that we ingest additional information via the model at the simulation and do not just naively smooth between scarce observations. We will make sure to emphasise this in the revised manuscript.

- L294: what do you really mean by: virtual variable?

We mean it in the sense of Beven et al. (2012).

- L296: what do you really mean by: is a new qualitative dimension?

With this we want to express that we still argue that simulations are qualitatively different from the streamflow "observations". Basically, the sentence makes the same argument as the first question/comment by the reviewer. We will make sure that this aspect will be more evident in the revised form of the manuscript.

- L296 "The job of the model is here to extract the information that the meteorological signals contain about the streamflow and act as a virtual sensor" is to me a vague statement.

We will rephrase this sentence to make it more clear. In the revised version we will write:

In our case, the model acts as a virtual sensor. It extracts the information that the meteorological signals contain about the streamflow. The ungauged EARLS basins do constitute a layer of information that one would not obtain from using streamflow observations only.

- L297-299: "This is certainly not a replacement for real data — on the contrary, our ability to enable models in this way heavily depends on the availability of large amounts of diverse and highly qualitative data (Kratzert et al., 2024). " I do not fully follow you here.

With this sentence we want to express that observational data is and will remain the most important factor for hydrological inquiry. We will formulate this clearer in the revised version of the manuscript.

---

## Author Comment (AC2)

**Reviewer: Juliane Mai**

Dear Daniel and others,

It was a pleasure to review your manuscript on "EARLS: A runoff reconstruction dataset for Europe", submitted to the Earth System Science Data journal. I found your study to be well-structured, informative, and a very valuable contribution to the field.

I have provided a list of minor comments for your consideration. Most of these do not require urgent revisions, but I have highlighted in blue those that may benefit from some additional attention.

My main comments focus on the following aspects:

- Slight inconsistencies between datasets – While these are not major issues and likely do not require changes, I would appreciate your expert opinion on what potential impacts these inconsistencies might have on your findings.

- Figure clarity – Some suggestions are provided to improve comparisons and ease interpretation.

- Simulation data and uncertainties – Comments regarding the data shared to ensure transparency and clarity.

Overall, I am recommending minor revisions. I appreciate the effort you have put into this work and would be happy to have another look at the revised version.

Best regards,

 Julie Mai

Dear Juliane,

Thank you for your extremely productive review. Your feedback was very detail oriented and wherever possible we implement the proposed changes for our revisions (see detailed comments). We think this will greatly improve the paper!

Detailed comments:

Section 1:

- None

Section 2:

- Line 74: "not able to produce a simulation for 227" —> Why? Are these 227 part of the 14,161 basins without data gaps, or the 2,655 with gaps, or additional ones?

For these basins we have so much missing data in the inputs that we were not able to model to make simulations at all. The 227 are not part of the 2,655. We will clarify this indeed.

- Line 83: "The basin shapes are either derived from EStreams (do Nascimento et al., 2024) or HydroATLAS (Linke et al., 2019)" —> For a given basin, what made you decide which one to use? If you specify this later please refer to that section here.

We used both whenever possible. We will make sure to mention this.

- Line 84: "We use 4 dynamic inputs" —> You state later that EOBS has more forcing variables. Are the other ones not contributing to a better performance? Did you test other variables? I think it would be great to have some additional information on this here.

The reviewer's intuition is completely right. The additional inputs can be used to improve model performance (albeit not by much). We chose a small amount of commonly available inputs to make it easy to apply the model for different use cases/scenarios. We will discuss this more in the revised manuscript.

- Line 88: "[EOBS is] spanning 1 January 1950 to present." —> Your dataset starts in 1953 though. Why are the first 3 years skipped. I think that's fine. It would just be good to have some information why you skipped 3 years.

This is indeed confusing for readers. We use the first three years as buffer years. We will mention this explicitly here.

- Line 90: "we use 13 attributes that we aggregate from HydroATLAS, namely: basin area, the average elevation, the average slopes, the average stream gradient, the average long-term air temperature, the minimum long-term air temperature, the maximum long- term air temperature, a global aridity index" —> I am wondering what happens if you are using a basin shape from EStreams (not HydroATLAS) to aggregate forcing etc but then you use basin characteristics that are (technically) for a (potentially) different basin shape. For example, basin area may be different but also meteorologically based basin characteristics would not be consistent. Further, why are the meteorologically based attributes to directly derived from the forcing dataset? I'm sure this is all just minor differences but some kind of note for the reader may be helpful.

This is an interesting question. If the introduced differences are systematic the LSTM could learn to compensate for them and there might not be a difference at all. If they would, however, have random components it would inject additional noise to the forcings. One can only guess how heavy the degradation in performance would then be. We will mention the possibility to create new EARLS version by recombining different datasets in the conclusions of the revised manuscript, together with the discussion of using different dataset all together for the statics (see reviewer comment regarding Table 1).

As for the meteorologically based attributes: For the static attributes we follow the convention of the Caravan dataset. The goal here was to make it so that, in theory, the model can be used for a wide range of situations. We will mention this explicitly in the revised manuscript.

- Figure 2: The positioning of the three-parameter Laplace distribution of simulation results just at the end of the LSTM time line looks bit like there is only one set of the three parameters estimated. It maybe clearer if each of the orange LSTM "boxes" would have an arrow (or connection) to the gamma, b,

and tau. Or maybe place an index "t" at each of the three parameters to show they are time dependent?

We agree that this depiction might be misleading. To remedy this we will add a time index to the figure (as proposed by the reviewer), and mention this explicitly in the caption.

- Line 146: "seperately delineated gauged basins" —> What does that mean? Basins have another shape than you used for EARLS? Doesn't that mean that the forcings are (slightly) different? What impact may it have that EARLS estimates are for a potentially slightly larger or smaller basin than mHM given that it is another basin shape? Also, typo: "seperately" —> "separately"

The formulation is indeed unfortunate. What we wanted to express is that they are not part of EARLS but are specifically used for the experiment. We will make sure to say this explicitly in the revised manuscript.

- Table 1: You list the datasets used to setup mHM. Technically these could be used to derive the static basin attributes for the LSTM. How different are estimates from those datasets compared to what you use from HydroATLAS? Would these differences matter?

This is an interesting question. We did some tests with the static attributes of EStreams. They are, generally speaking, of higher quality than the ones from HydroATLAS, but secluded to Eruope. Roughly speaking, what we saw there is that the ungauged basin performance increased a little. In order to be able to use the first EARLS LSTM to a wider area we did, however, decide to use the static attributes that are globally available. That said, the broader idea here is perhaps to build different EARLS versions with other static attributes in the future. We will make sure to discuss this in the conclusions of the revised manuscript

Section 3:

- "potentially difficult to predict ungauged basins" —> It is stated that all basins have "natural" streamflow (line 53). I was hence expecting that these are mostly pristine basins with low human impact. Is that correct? What

would make these basins then "potentially difficult to predict"? It would be great if some reasons why a basin may be "potentially difficult to predict" would be added to the manuscript. Also, out of curiosity, how do you know that basins have "natural" streamflow. Is that a HydroATLAS attribute for basins? If so, maybe mention that?

The statement of the reviewer is correct. We use the pre-filtering described in appendix A1 (However, this still leaves a lot of ungauged basins that can be difficult to handle, e.g., Karst-affected catchments, anthropogenic influences, etc.). This specific sentence, however, specifically refers to the data splitting approach that we describe in appendix A3. That is, we do not split the data completely randomly, but make it so that the train-, validation-, test-data exhibit distributional shifts. The way we formulated this does indeed make this unclear. In the revised manuscript, we will make this clearer and refer explicitly to appendix A3.

- Line 169: I think it would be helpful for the reader if you could start the section with repeating the time period you evaluate here.

We will do so in the revised manuscript.

- Line 171: suggestion: "opposed to Kratzert et al., 2021" —> "opposed to Kratzert et al., 2021 using multiple facing datasets"

We will change this as proposed.

- Line 172: "These results are similar to the ones in Kratzert et al. (2019a), but worse than the ones from Mai et al. (2022)." —> Would it be possible to state how many where below 0.0 in those two studies? It would help for comparison.

If this is helpful we will include the specific information in the revised manuscript. Specifically: TK

- Figure 3: The validation data line is barely visible. Maybe use thinner lines for all three and remove (some) gridlines? It would also be nice to state clearly

how many basins are in each of the three sets. Maybe add to the legend (e.g., "test data (N=500)")?

We agree with the analysis. Of course the fact that all lines are closely alone is part of the argument. Still more information is good for readers here. Hence, we will adapt this as proposed by the reviewer.

- Figure 4:
    - The caption states that "The colors mirror Fig. 3". But figure 3 has colours for test, training and validation data. Figure 4 shows NSE performances of the validation (?) data.
    - I think figure 3 and 4 should be combined since they are somewhat connected. The continuous colourbar in figure 4 is not really needed I think. I can only distinguish 3 colours- blue, red and purple. I would just pick 3-4 distinct colours (or the ones you use in for example Figure 8) and then use those to get the point across. Also, I think the colourbar needs to be open ended at the lower end as I am assuming that there is values smaller than -0.5 that are also coloured in red?!
    - It would be great to exactly state which basins are in figure 4. I am guessing validation basins. I would use the same exact wording as used in figure 3.

We did indeed mess this up. We will adapt this as proposed by the reviewer.

- Line 175-190: I really like this discussion. It states a lot about the methodology (data pre-processing and filtering of basins) that I was expecting beforehand (see comments above). I think it maybe advisable to have a separate section called "data pre-processing" where all this is placed before we dive into model performance etc.

If the editor allows for it we will adapt this as proposed by the reviewer.

- Figure 5: caption suggestion:
    - "Bertola distances" —> "Bertola distances d"
    - Please add to the caption what panel a and b depicts.

- Typo: "D_{<0.8}" —> "D_{>0.8}" (in caption)

We will adapt this as proposed by the reviewer.

- Line 199: "Tn" —> "In"

Thank you.

- Line 201: "roughly starting at the 60th percentile" —> I would actually say much earlier; like around 37.5th percentile. Unless the authors look at another feature or criterion to determine when models start looking similar in performance… It maybe helpful to have those lines added as horizontal lines in Figure 6 for the reader to better be directed towards what they are supposed to see.

We will adapt this as proposed by the reviewer.

- Line 201: "For the remaining 15%" —> I am not sure which remaining 15%.
  - 0-15th = EARLS and default are similar
  - 60-100th (I suggest 35-100th)= EARLS and local are similar
  - 15-60th (I suggest 15-35th) = in between -> but this is 45% (unless the authors agree that it should be around 20% remaining)

Exactly. As with the comment before, we will adapt the wording of the reviewer in the revised manuscript.

- Figure 7: I love that figure!

Thank you!

- Line 207: "We encourage readers to compare our version with these depictions" —> Please be aware that the publication is not open access. So, it may limit the ability of readers to actually do this. :( I am assuming it is not reasonable to recreate the figure with the Blöschl data and have them for comparison in the manuscript?

Unfortunately this is exactly the case.

- Figure 8: The caption is not consistent with the colorbar. "Positive trends are depicted in red" but they are blue and vice versa. Unless the legend shows Blöschl minus EARLS estimates and then you talk about EARLS minus Blöschl in the figure caption. In any case it's confusing.

We will correct this.

- Line 219: "the original version shows slightly positive trends" —> Isn't figure 9a all "red" in Scandinavia which is negative values which means negative trends (see caption figure 9). There is some sort of mix-up what positive and negative means I think. There maybe more in this paragraph but I leave it to the authors to revise them without pointing each one out here.

Yes. This is indeed a mixup leaching from figure 8. We will correct this together with your comment on Figure 9.

- Line 219: "positive trends" and "negative trends" —> in general, I am not sure if I would refer to them as "positive" and "negative". I am understanding that positive values indicate more floods and I am not sure if that's "positive". Maybe refer to them as "trends of increasing number of floods" or "increasing trend of floods" or something…

We will adapt the description as proposed.

- Line 224: "Some differences can be explained by data availability" —> Do you think it may be helpful to actually plot the gauge stations used for the two datasets in figures 9a and 9b? It may underline your point that differences appear where more data are available while the other dataset lacks observations.

Yes. This comment is related to comment TK by Wouther Berghuis. We do indeed agree that contextualizing the nature of the differences makes the manuscript clearer and more readable. Hence, we will include the proposed solution in the revised manuscript.

- Figure 9: Would it be possible to use the same colorbar as Figure 8?

Yes.

- Figure A1 and A2: I highly recommend to merge these two figures into one figure with two panels. This would make it much easier to compare the workflows; especially when the box-diagrams (which are beautiful) are arranged in the same way.

We will adapt the figure as proposed in the comment.

> I think the "selection" step in either figure needs to be part of the methodology. Currently this is a bit vague and distributed. It's such a curial step that it should be easily findable. I later found some of that in the Appendix but maybe move it to the methods or at least refer to this section of the appendix early on in the methods.

Albeit this is an attractive proposal we don't think that we can do that. As a matter of fact, the figures were part of the method section for the first version of the manuscript. However, the editor asked us to move them to the appendix to align the manuscript more with ESSD. Hence, if the editor does not ask us explicitly to move them back up, we will keep them as part of the appendix.

> Also, there should be a comparable section like this for the section of "engaged" basins, right?

We will include such a section.

- Table B1: Wow!

All accolades go to Thiago for this one!

- Figure C1 to C3: I think these three can be merged into one figure with three panels. It would be easier to compare them if they are next to each other. Also, the KGE could just run from -2 or -1 to 1. This way one would see more of the actual interesting part of the rising limb.

We will adapt these changes for the revised manuscript.

Section 4:

- Line 256: You may want to link to DOI "10.5281/zenodo.13864842" which would always point to the latest version of the dataset.

We will do that.

- Line 263: The constructions folder is stated to include CSV files with at least date and simulation in mm/day. That is great. I am however wondering if it would be possible to include some information about the uncertainty estimates. I think these are a major selling point of this dataset and it is stated that it is included later (line 289-291: "each time step EARLS provides a conditional uncertainty estimate — which can, for example, be used to compute the likelihood of a given model" (which I really like). I was however not able to download the full 33GB dataset and check if there may be a file that contains the uncertainty information. The estimated download time was 26 hours which seemed too much to wait…
    - If the uncertainty data are contained, please make more clear where one would find these data.

The uncertainty data are indeed part of the dataset. We will describe this more thoroughly in the revised version.

    - If it's not included maybe make more clear that a user would need to setup an LSTM themselves and train it and then get those estimates themselves.

The LSTM is not provided as part of the dataset, but we will provide the model structure and weights it as part of the code of the paper. The revised manuscript will emphasize this in the code section.

    - Is the download always taking so long or is it just me? An idea would be to have a mini-example with 3-5 basins in a separate (much smaller) zip such that people could download that to see if it contains what they would expect, and setup workflows while they wait for the entire pack to download?! Up to the authors, of course.

We will do exactly that! When we tried it it did not take that long. But as the reviewer points out there can always be circumstances that lead to slower download times. Providing a small sample of the dataset solves this. Great!

Section 5:

- Line 288: "11 thousand" —> "11,000"

Acknowledgements:

- Line 433: "mHm" —> "mHM"

Thank you.